# PAC-Bayes Un-Expected Bernstein Inequality

**Zakaria Mhammedi**
The Australian National University and Data61
zak.mhammedi@anu.edu.au

**Peter D. Grünwald**
CWI and Leiden University
pdg@cwi.nl

**Benjamin Guedj**
Inria and University College London
benjamin.guedj@inria.fr

## Abstract

We present a new PAC-Bayesian generalization bound. Standard bounds contain a $\sqrt{L_n \cdot \mathrm{KL}/n}$ complexity term which dominates unless $L_n$, the empirical error of the learning algorithm's randomized predictions, vanishes. We manage to replace $L_n$ by a term which vanishes in many more situations, essentially whenever the employed learning algorithm is sufficiently stable on the dataset at hand. Our new bound consistently beats state-of-the-art bounds both on a toy example and on UCI datasets (with large enough $n$). Theoretically, unlike existing bounds, our new bound can be expected to converge to $0$ faster whenever a Bernstein/Tsybakov condition holds, thus connecting PAC-Bayesian generalization and *excess risk* bounds—for the latter it has long been known that faster convergence can be obtained under Bernstein conditions. Our main technical tool is a new concentration inequality which is like Bernstein's but with $X^2$ taken outside its expectation.

## 1 Introduction

PAC-Bayesian generalization bounds [1, 8, 9, 17, 18, 20, 28, 29, 30] have recently obtained renewed interest within the context of deep neural networks [14, 34, 42]. In particular, Zhou et al. [42] and Dziugaite and Roy [14] showed that, by extending an idea due to Langford and Caruana [23], one can obtain nontrivial (but still not very strong) generalization bounds on real-world datasets such as MNIST and ImageNet. Since using alternative methods, nontrivial generalization bounds are even harder to get, there remains a strong interest in improved PAC-Bayesian bounds. In this paper, we provide a considerably improved bound whenever the employed learning algorithm is sufficiently *stable* on the given data.

Most standard bounds have an order $\sqrt{L_n \cdot \mathrm{COMP}_n/n}$ term on the right, where $\mathrm{COMP}_n$ represents model complexity in the form of a Kullback-Leibler divergence between a prior and a posterior, and $L_n$ is the *posterior expected loss* on the training sample. The latter only vanishes if there is a sufficiently large neighborhood around the "center" of the posterior at which the training error is $0$. In the two papers [14, 42] mentioned above, this is not the case. For example, the various deep net experiments reported by Dziugaite et al. [14, Table 1] with $n = 150000$ all have $L_n$ around $0.03$, so that $\sqrt{\mathrm{COMP}_n/n}$ is multiplied by a non-negligible $\sqrt{0.03} \approx 0.17$. Furthermore, they have $\mathrm{COMP}_n$ increasing substantially with $n$, making $\sqrt{L_n \cdot \mathrm{COMP}_n/n}$ converge to $0$ at rate slower than $1/\sqrt{n}$.

In this paper, we provide a bound (Theorem 3) with $L_n$ replaced by a second-order term $V_n$—a term which will go to $0$ in many cases in which $L_n$ does not. This can be viewed as an extension of an earlier second-order approach by Tolstikhin and Seldin [39] (TS from now on); they also replace $L_n$, but by a term that, while usually smaller than $L_n$, will tend to be larger than our $V_n$. Specifically, as

they write, in classification settings (our primary interest), their replacement is not much smaller than $L_n$ itself. Instead our $V_n$ can be very close to 0 in classification even when $L_n$ is large. While the TS bound is based on an "empirical" Bernstein inequality due to [27][1], our bound is based on a different modification of Bernstein's moment inequality in which the occurrence of $X^2$ is taken outside of its expectation (see Lemma 13). We note that an empirical Bernstein inequality was introduced in [4, Theorem 1], and the name "Empirical Bernstein" was coined in [32].

The term $V_n$ in our bound goes to 0—and our bound improves on existing bounds—whenever the employed learning algorithm is relatively stable on the given data; for example, if the predictor learned on an initial segment (say, $50\%$) of the dataset performs similarly (*i.e.* assigns similar losses to the same samples) to the predictor based on the full data. This improvement is reflected in our experiments where, except for very small sample sizes, we consistently outperform existing bounds both on a toy classification problem with label noise and on standard UCI datasets [13]. Of course, the importance of stability for generalization has been recognized before in landmark papers such as [7, 33, 38], and recently also in the context of PAC-Bayes bounds [35]. However, the data-dependent stability notion "$V_n$" occurring in our bound seems very different from any of the notions discussed in those papers.

Theoretically, a further contribution is that we connect our PAC-Bayesian generalization bound to *excess risk bounds*; we show that (Theorem 7) our generalization bound can be of comparable size to excess risk bounds up to an irreducible *complexity-free* term that is independent of model complexity. The excess risk bound that can be attained for any given problem depends both on the complexity of the set of predictors $\mathcal{H}$ and on the inherent "easiness" of the problem. The latter is often measured in terms of the exponent $\beta \in [0, 1]$ of the *Bernstein condition* that holds for the given problem [6, 15, 19], which generalizes the exponent in the celebrated *Tsybakov margin condition* [5, 40]. The larger $\beta$, the faster the excess risk converges. In Section 5, we essentially show that the rate at which the $\sqrt{V_n \cdot \mathrm{COMP}_n / n}$ term goes to 0 can also be bounded by a quantity that gets smaller as $\beta$ gets larger. In contrast, previous PAC-Bayesian bounds do not have such a property.

**Contents.** In Section 2, we introduce the problem setting and provide a first, simplified version of our main theorem. Section 3 gives our main bound. Experiments are presented in Section 4, followed by theoretical motivation in Section 5. The proof of our main bound is provided in Section 6, where we first present the convenient ESI language for expressing stochastic inequalities, and (our main tool) the unexpected Bernstein lemma (Lemma 13). The paper ends with an outlook for future work.

## 2   Problem Setting, Background, and Simplified Version of Our Bound

**Setting and Notation.**   Let $Z_1, \ldots, Z_n$ be i.i.d. random variables in some set $\mathcal{Z}$, with $Z_1 \sim \mathbf{D}$. Let $\mathcal{H}$ be a hypothesis set and $\ell : \mathcal{H} \times \mathcal{Z} \to [0, b]$, $b > 0$, be a bounded loss function such that $\ell_h(Z) \coloneqq \ell(h, Z)$ denotes the loss that hypothesis $h$ makes on $Z$. We call any such tuple $(\mathbf{D}, \ell, \mathcal{H})$ a *learning problem*. For a given hypothesis $h \in \mathcal{H}$, we denote its *risk* (expected loss on a test sample of size 1) by $L(h) \coloneqq \mathbb{E}_{Z \sim \mathbf{D}} [\ell_h(Z)]$ and its empirical error by $L_n(h) \coloneqq \frac{1}{n} \sum_{i=1}^n \ell_h(Z_i)$. For any distribution $P$ on $\mathcal{H}$, we write $L(P) \coloneqq \mathbb{E}_{h \sim P} [L(h)]$ and $L_n(P) \coloneqq \mathbb{E}_{h \sim P} [L_n(h)]$.

For any $m \in [n]$ and any variables $Z_1, \ldots, Z_n$ in $\mathcal{Z}$, we denote $Z_{\leq m} \coloneqq (Z_1, \ldots, Z_m)$ and $Z_{<m} \coloneqq Z_{\leq m-1}$, with the convention that $Z_{\leq 0} = \varnothing$. Similarly, we denote $Z_{\geq m} \coloneqq (Z_m, \ldots, Z_n)$ and $Z_{>m} \coloneqq Z_{\geq m+1}$, with the convention that $Z_{\geq n+1} = \varnothing$. As is customary in PAC-Bayesian works, a *learning algorithm* is a (computable) function $P : \bigcup_{i=1}^n \mathcal{Z}^i \to \mathcal{P}(\mathcal{H})$ that, upon observing input $Z_{\leq n} \in \mathcal{Z}^n$, outputs a "posterior" distribution $P(Z_{\leq n})(\cdot)$ on $\mathcal{H}$. The posterior could be a Gibbs or a generalized-Bayesian posterior but also other algorithms. When no confusion can arise, we will abbreviate $P(Z_{\leq n})$ to $P_n$, and denote $P_0$ any "prior" distribution, *i.e.* a distribution on $\mathcal{H}$ which has to be specified in advance, before seeing the data; we will use the convention $P(\varnothing) = P_0$. Finally, we denote the Kullback-Leibler divergence between $P_n$ and $P_0$ by $\mathrm{KL}(P_n \| P_0)$.

**Comparing Bounds.** Both existing state-of-the-art PAC-Bayes bounds and ours essentially take the following form; there exists constants $\mathcal{P}, \mathcal{A}, \mathcal{C} \geq 0$, and a function $\varepsilon_{\delta,n}$, logarithmic in $1/\delta$ and $n$, such

that for all $\delta \in ]0, 1[$, with probability at least $1 - \delta$ over the sample $Z_1, \ldots, Z_n$, it holds that,

$$L(P_n) - L_n(P_n) \leq \mathcal{P} \cdot \sqrt{\frac{R_n \cdot (\text{COMP}_n + \varepsilon_{\delta,n})}{n}} + \mathcal{A} \cdot \frac{\text{COMP}_n + \varepsilon_{\delta,n}}{n} + \mathcal{C} \cdot \sqrt{\frac{R'_n \cdot \varepsilon_{\delta,n}}{n}}, \quad (1)$$

where $R_n, R'_n \geq 0$ are sample-dependent quantities which may differ from one bound to another. Existing classical bounds that after slight relaxations take on this form are due to Langford and Seeger [24, 37], Catoni [10], Maurer [26], and Tolstikhin and Seldin (TS) [39] (see the latter for a nice overview). In all these cases, $\text{COMP}_n = \text{KL}(P_n \| P_0)$, $R'_n = 0$, and—except for the TS bound—$R_n = L_n(P_n)$. For the TS bound, $R_n$ is equal to the empirical loss variance. Our bound in Theorem 3 also fits (1) (after a relaxation), but with considerably different choices for $\text{COMP}_n$, $R'_n$, and $R_n$.

Of special relevance in our experiments is the bound due to Maurer [26], which as noted by TS [39] tightens the PAC-Bayes-kl inequality due to Seeger [36], and is one of the tightest known generalization bounds in the literature. It can be stated as follows: for $\delta \in ]0, 1[$, $n \geq 8$, and any learning algorithm $P$, with probability at least $1 - \delta$,

$$\text{kl}(L(P_n), L_n(P_n)) \leq \frac{\text{KL}(P_n \| P_0) + \ln \frac{2\sqrt{n}}{\delta}}{n}, \quad (2)$$

where kl is the binary Kullback-Leibler divergence. Applying the inequality $p \leq q + \sqrt{2q \, \text{kl}(p\|q)} + 2 \, \text{kl}(p\|q)$ to (2) yields a bound of the form (1) (see [39] for more details). Note also that using Pinsker's inequality together with (2) implies McAllester's classical PAC-Bayesian bound [28].

We now present a simplified version of our bound in Theorem 3 below as a corollary.

**Corollary 1.** *For any $1 \leq m < n$ and any deterministic estimator $\hat{h} : \bigcup_{i=1}^{n} \mathcal{Z}^i \to \mathcal{H}$ (such as ERM), there exists $\mathcal{P}, \mathcal{A}, \mathcal{C} > 0$, such that (1) holds with probability at least $1 - \delta$, with*

$$\text{COMP}_n = \text{KL}(P_n \| P(Z_{\leq m})) + \text{KL}(P_n \| P(Z_{>m})),$$

$$R'_n := V'_n := \frac{1}{n} \sum_{i=1}^{m} \ell_{\hat{h}(Z_{>m})}(Z_i)^2 + \frac{1}{n} \sum_{j=m+1}^{n} \ell_{\hat{h}(Z_{\leq m})}(Z_j)^2, \quad (3)$$

$$R_n := V_n := \frac{1}{n} \mathbb{E}_{h \sim P_n} \left[ \sum_{i=1}^{m} \left( \ell_h(Z_i) - \ell_{\hat{h}(Z_{>m})}(Z_i) \right)^2 + \sum_{j=m+1}^{n} \left( \ell_h(Z_j) - \ell_{\hat{h}(Z_{\leq m})}(Z_j) \right)^2 \right].$$

Like in TS's and Catoni's bound, but unlike McAllester's and Maurer's, our $\varepsilon_{\delta,n}$ grows as $(\ln \ln n)/\delta$. Another difference is that our complexity term is a sum of two KL divergences, in which the prior (in this case $P(Z_{\leq m})$ or $P(Z_{>m})$) is "informed"—when $m = n/2$, it is really the posterior based on half the sample. Our experiments confirm that this tends to be much smaller than $\text{KL}(P_n \| P_0)$. While the idea to use part of the sample to create an informed prior is due to [2], we are the first to combine all parts (halves) into a single bound, which requires a novel technique. This technique can be applied to other existing bounds as well (see Section 3).

A larger difference between our bound and others is in the fact that we have $R_n = V_n$ instead of the typical empirical error $R_n = L_n(P_n)$. Only TS [39] have a $R_n$ that is somewhat reminiscent of ours; in their case $R_n = \mathbb{E}_{h \sim P_n} [\sum_{i=1}^{n} (\ell_h(Z_i) - L_n(h))^2]/(n-1)$ is the empirical loss variance. The crucial difference to our $V_n$ is that the empirical loss variance cannot be close to 0 unless a sizeable $P_n$-posterior region of $h$ has empirical error almost constant on most data instances. For classification with 0-1 loss, this is a strong condition since the empirical loss variance is equal to $nL_n(P_n)(1 - L_n(P_n))/(n-1)$, which is only close to 0 if $L_n(P_n)$ is itself close to 0 or 1. In contrast, our $V_n$ can go to zero 0 even if the empirical error and variance do not, as long as the learning algorithm is sufficiently stable. This can be witnessed in our experiments in Section 4. In Section 5, we argue more formally that under a Bernstein condition, the $\sqrt{V_n \cdot \text{COMP}_n/n}$ term in our bound can be much smaller than $\sqrt{\text{COMP}_n/n}$. Note, finally, that the term $V_n$ has a two-fold cross-validation flavor, but in contrast to a cross-validation error, for $V_n$ to be small, it is sufficient that the losses are *similar*, not that they are small.

The price we pay for having $R_n = V_n$ in our bound is the right-most, irreducible remainder term in (1) of order at most $b/\sqrt{n}$. Note, however, that this term is decoupled from the complexity $\text{COMP}_n$, and thus it is not affected by $\text{COMP}_n$ growing with the "size" of $\mathcal{H}$. The following lemma gives a tighter bound (tighter than the $b/\sqrt{n}$ just mentioned) on the irreducible term:

**Lemma 2.** *Suppose that the loss is bounded by 1 (*i.e. $b = 1$*) and that $n$ is even, and let $m = n/2$. For $\delta \in ]0, 1[$, $R'_n$ as in (3), and any estimator $\hat{h} : \bigcup_{i=1}^{n} \mathcal{Z}^i \to \mathcal{H}$, we have, with probability at least $1 - \delta$,*

$$\sqrt{\frac{R'_n}{n}} \leq \sqrt{\frac{2(L(\hat{h}(Z_{>m})) + L(\hat{h}(Z_{\leq m})))}{n}} + \frac{4\sqrt{\ln \frac{4}{\delta}}}{n}. \tag{4}$$

Behind the proof of the lemma is an application of Hoeffding's and the empirical Bernstein inequality [27] (see Section C). Note that in the realizable setting, the first term on the RHS of (4) can be of order $O(1/n)$ with the right choice of estimator $\hat{h}$ (*e.g.* ERM). In this case (still in the realizable setting), our irreducible term would go to zero at the same rate as other bounds which have $R_n = L_n(P_n)$.

## 3 Main Bound

We now present our main result in its most general form. Let $\vartheta(\eta) := (-\ln(1 - \eta) - \eta)/\eta^2$ and $c_\eta := \eta \cdot \vartheta(\eta b)$, for $\eta \in ]0, 1/b[$, where $b > 0$ is an upper-bound on the loss $\ell$.

**Theorem 3. [Main Theorem]** *Let $Z_1, \ldots, Z_n$ be i.i.d. with $Z_1 \sim \mathbf{D}$. Let $m \in [0..n]$ and $\pi$ be any distribution with support on a finite or countable grid $\mathcal{G} \subset ]0, 1/b[$. For any $\delta \in ]0, 1[$, and any learning algorithms $P, Q : \bigcup_{i=1}^{n} \mathcal{Z}^i \to \mathcal{P}(\mathcal{H})$, we have,*

$$L(P_n) \leq L_n(P_n) + \inf_{\eta \in \mathcal{G}} \left\{ c_\eta \cdot V_n + \frac{\text{COMP}_n + 2\ln \frac{1}{\delta \cdot \pi(\eta)}}{\eta \cdot n} \right\} + \inf_{\nu \in \mathcal{G}} \left\{ c_\nu \cdot V'_n + \frac{\ln \frac{1}{\delta \cdot \pi(\nu)}}{\nu \cdot n} \right\}, \tag{5}$$

*with probability at least $1 - \delta$, where $\text{COMP}_n$, $V'_n$, and $V_n$ are the random variables defined by:*

$$\text{COMP}_n := \text{KL}(P_n \| P(Z_{\leq m})) + \text{KL}(P_n \| P(Z_{>m})), \tag{6}$$

$$V'_n := \frac{1}{n} \sum_{i=1}^{m} \mathbb{E}_{h \sim Q(Z_{>i})} \left[ \ell_h(Z_i)^2 \right] + \frac{1}{n} \sum_{j=m+1}^{n} \mathbb{E}_{h \sim Q(Z_{<j})} \left[ \ell_h(Z_j)^2 \right],$$

$$V_n := \frac{1}{n} \mathbb{E}_{h \sim P_n} \left[ \sum_{i=1}^{m} \left( \ell_h(Z_i) - \mathbb{E}_{h' \sim Q(Z_{>i})} \left[ \ell_{h'}(Z_i) \right] \right)^2 + \sum_{j=m+1}^{n} \left( \ell_h(Z_j) - \mathbb{E}_{h' \sim Q(Z_{<j})} \left[ \ell_{h'}(Z_j) \right] \right)^2 \right].$$

While the result holds for all $0 \leq m \leq n$, in the remainder of this paper, we assume for simplicity that $n$ is even and that $m = n/2$. We will also be using the grid $\mathcal{G}$ and distribution $\pi$ defined by

$$\mathcal{G} := \left\{ \frac{1}{2b}, \ldots, \frac{1}{2^K b} : K := \left\lceil \log_2 \left( \frac{1}{2} \sqrt{\frac{n}{\ln \frac{1}{\delta}}} \right) \right\rceil \right\}, \quad \text{and} \quad \pi \equiv \text{uniform distribution over } \mathcal{G}. \tag{7}$$

Roughly speaking, this choice of $\mathcal{G}$ ensures that the infima in $\eta$ and $\nu$ in (5) are attained within $[\min \mathcal{G}, \max \mathcal{G}]$. Using the relaxation $c_\eta \leq \eta/2 + \eta^2 11b/20$, for $\eta \leq 1/(2b)$, in (5) and tuning $\eta$ and $\nu$ within the grid $\mathcal{G}$ defined in (7) leads to a bound of the form (1). Furthermore, we see that the expression of $V_n$ in Corollary 1 now follows when $Q$ is chosen such that, for $1 \leq i \leq m < j \leq n$, $Q(Z_{>i}) \equiv \delta(\hat{h}(Z_{>m}))$ and $Q(Z_{<j}) \equiv \delta(\hat{h}(Z_{\leq m}))$, for some deterministic estimator $\hat{h}$, where $\delta(h)(\cdot)$ denotes the Dirac distribution at $h \in \mathcal{H}$.

**Online Estimators.** It is clear that Theorem 3 is considerably more general than its Corollary 1; when predicting the $j$-th point $Z_j$, $j > m$, in the RHS sum of $V_n$, we could use a posterior $Q(Z_{<j}) \equiv \delta(\hat{h}(Z_{<j}))$ which does not only depend on $Z_1, \ldots, Z_m$, but also on part of the second sample, namely $Z_{m+1}, \ldots, Z_{j-1}$, and analogously when predicting $Z_i$, $i \leq m$, in the LHS sum of $V_n$. We can thus base our bound on a sum of errors achieved by *online estimators* $(\hat{h}(Z_{<j}))$ and $(\hat{h}(Z_{>i}))$ which converge to the final $\hat{h}(Z_{\leq n})$ based on the full data. Doing this would likely improve our bounds, but we did not try it in our experiments since it is computationally demanding.

**Informed Priors.** Other bounds can also be modified to make use of "informed priors" from each half of the data; in this case, the $\text{KL}(P_n \| P_0)$ term in these bounds can be replaced by $\text{COMP}_n$ defined in (6). As revealed by additional experiments in the Appendix H, doing this substantially improves the corresponding bounds when the learning algorithm is sufficiently stable. Here we show how this can be done for Maurer's bound in (2) (the details for other bounds are postponed to Appendix A).

**Lemma 4.** *Let $\delta \in ]0,1[$ and $m \in [0..n]$. In the setting of Theorem 3, we have, with probability at least $1 - \delta$,*

$$\mathrm{kl}(L(P_n), L_n(P_n)) \leq \frac{\mathrm{KL}(P_n\|P(Z_{\leq m})) + \mathrm{KL}(P_n\|P(Z_{>m})) + \ln\frac{4\sqrt{m(n-m)}}{\delta}}{n}.$$

**Remark 5.** *(Useful for Section 5 below) Though this may deteriorate the bound in practice, Theorem 3 allows choosing a learning algorithm $P$ such that for $1 \leq m < n$, $P(Z_{\leq m}) \equiv P(Z_{>m}) \equiv P_0$ (i.e. no informed priors); this results in $\mathrm{COMP}_n = 2\mathrm{KL}(P_n\|P_0)$—the bound is otherwise unchanged.*

**Biasing.** The term $V_n$ in our bound can be seen as the result of "biasing" the loss when evaluating the generalization error on each half of the sample. The TS bound, having a second order variance term, can be used in a way as to arrive at a bound like ours with the same $V_n$ as in Corollary 1. The idea here is to apply the TS bound twice (once on each half of the sample) to the biased losses $\ell(h,\cdot) - \ell(\hat{h}(Z_{\leq m}),\cdot)$ and $\ell(h,\cdot) - \ell(\hat{h}(Z_{>m}),\cdot)$, then combine the results with a union bound. The details of this are postponed to Appendix B. Note however, that this trick will not lead to a bound with a $V_n$ term as in Theorem 3, *i.e.* with the online posteriors $(Q(Z_{>i}))$ and $(Q(Z_{<j}))$ which get closer and closer to the final $Q(Z_{\leq m})$ based on the full sample.

## 4 Experiments

In this section, we experimentally compare our bound in Theorem 3 to that of TS [39], Catoni [9, Theorem 1.2.8] (with $\alpha = 2$), and Maurer in (2). For the latter, given $L_n(P_n) \in [0,1[$ and the RHS of (2), we solve for an upper bound of $L(P_n)$ by "inverting" the kl. We note that TS [39] do not claim that their bound is better than Maurer's in classification (in fact, they do better in other settings).

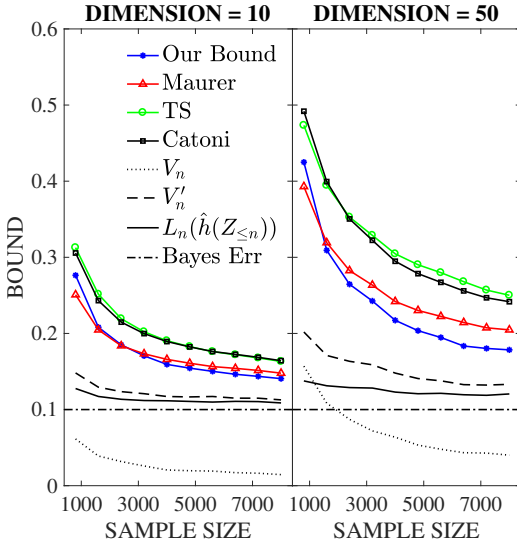

Figure 1: Results for the synthetic data.

| UCI Dataset | d | Test err. of $\hat{h}$ | Our bound | Maurer bound |
|---|---|---|---|---|
| Haberman ($n$=244) | 3 | 0.272 | 0.521 | **0.411** |
| Breast-C. ($n$=560) | 9 | 0.068 | 0.185 | **0.159** |
| TicTacToe ($n$=766) | 27 | 0.046 | **0.191** | 0.216 |
| Banknote ($n$=1098) | 4 | 0.058 | **0.125** | 0.136 |
| kr-vs-kp ($n$=2556) | 73 | 0.044 | **0.108** | 0.165 |
| Spambase ($n$=3680) | 57 | 0.173 | **0.293** | 0.312 |
| Mushroom ($n$=6500) | 116 | 0.002 | **0.018** | 0.055 |
| Adult ($n$=24130) | 108 | 0.168 | **0.195** | 0.234 |

Table 1: Results for the UCI datasets.

**Setting.** We consider both synthetic and real-world datasets for binary classification, and we evaluate bounds using the 0-1 loss. In particular, the data space $\mathcal{Z}$ is $\mathcal{X} \times \mathcal{Y} \coloneqq \mathbb{R}^d \times \{0,1\}$, where $d \in \mathbb{N}$ is the dimension of the feature space. In this case, the hypothesis set $\mathcal{H}$ is also $\mathbb{R}^d$, and the error associated with $h \in \mathcal{H}$ on a sample $Z = (X,Y) \in \mathcal{X} \times \mathcal{Y}$ is given by $\ell_h(Z) = |Y - \mathbb{1}\{\phi(h^\top X) > 1/2\}|$, where $\phi(w) \coloneqq 1/(1 + e^{-w}), w \in \mathbb{R}$. We learn our hypotheses using *regularized logistic regression*; given a sample $S = (Z_p, \ldots, Z_q)$, with $(p,q) \in \{(1,m),(m+1,n),(1,n)\}$ and $m = n/2$, we compute

$$\hat{h}(S) \coloneqq \arg\min_{h\in\mathcal{H}} \frac{\lambda\|h\|^2}{2} + \frac{1}{q-p+1}\sum_{i=p}^{q} Y_i \cdot \ln\phi(h^\top X_i) + (1-Y_i) \cdot \ln(1 - \phi(h^\top X_i)). \quad (8)$$

For $Z_{\leq n} \in \mathcal{Z}^n$, and $1 \leq i \leq m < j \leq n$, we choose algorithm $Q$ in Theorem 3 such that

$$Q(Z_{>i}) \equiv \delta\left(\hat{h}(Z_{>m})\right) \quad \text{and} \quad Q(Z_{<j}) \equiv \delta(\hat{h}(Z_{\leq m})).$$

Given a sample $S \neq \varnothing$, we set the "posterior" $P(S)$ to be a Gaussian centered at $\hat{h}(S)$ with variance $\sigma^2 > 0$; that is, $P(S) \equiv \mathcal{N}(\hat{h}(S), \sigma^2 I_d)$. The prior distribution is set to $P_0 \equiv \mathcal{N}(0, \sigma_0^2 I_d)$, for $\sigma_0 > 0$.

**Parameters.** We set $\delta = 0.05$. For all datasets, we use $\lambda = 0.01$, and (approximately) solve (8) using the BFGS algorithm. For each bound, we pick the $\sigma^2 \in \{1/2, \ldots, 1/2^J : J \coloneqq \lceil \log_2 n \rceil\}$ which minimizes it on the given data (with $n$ instances). In order for the bounds to still hold with probability at least $1 - \delta$, we replace $\delta$ on the RHS of each bound by $\delta/\lceil \log_2 n \rceil$ (this follows from the application of a union bound). We choose the prior variance such that $\sigma_0^2 = 1/2$ (this was the best value on average for the bounds we compare against). We choose the grid $\mathcal{G}$ in Theorem 3 as in (7). Finally, we approximate Gaussian expectations using Monte Carlo sampling.

**Synthetic data.** We generate synthetic data for $d = \{10, 50\}$ and sample sizes between 800 and 8000. For a given sample size $n$, we 1) draw $X_1, \ldots, X_n$ [resp. $\epsilon_1, \ldots, \epsilon_n$] identically and independently from the multivariate-Gaussian distribution $\mathcal{N}(0, I_d)$ [resp. the Bernoulli distribution $\mathcal{B}(0.9)$]; and 2) we set $Y_i = \mathbb{1}\{\phi(h_*^\top X_i) > 1/2\} \cdot \epsilon_i$, for $i \in [n]$, where $h_* \in \mathbb{R}^d$ is the vector constructed from the first $d$ digits of $\pi$. For example, if $d = 10$, then $h_* = (3, 1, 4, 1, 5, 9, 2, 6, 5, 3)^\top$. Figure 1 shows the results averaged over 10 independent runs for each sample size.

**UCI datasets.** For the second experiment, we use several UCI datasets. These are listed in Table 1 (where Breast-C. stands for Breast Cancer). We encode categorical variables in appropriate 0-1 vectors. This effectively increases the dimension of the input space (this is reported as $d$ in Table 1). After removing any rows (*i.e.* instances) containing missing features and performing the encoding, the input data is scaled such that every column has values between -1 and 1. We used a 5-fold train-test split ($n$ in Table 1 is the training set size), and the results in Table 1 are averages over 5 runs. We only compare with Maurer's bound since other bounds were worse than Maurer's and ours on all datasets.

**Discussion.** As the dimension $d$ of the input space increases, the complexity $\mathrm{KL}(P_n \| P_0)$—and thus, all the PAC-Bayes bounds discussed in this paper—get larger. Our bound suffers less from this increase in $d$, since for a large enough sample size $n$, the term $V_n$ is small enough (see Figure 1) to absorb any increase in the complexity. In fact, for large enough $n$, the irreducible (complexity-free) term involving $V_n'$ in our bound becomes the dominant one. This, combined with the fact that for the 0-1 loss, $V_n' \approx L_n(P_n)$ for large enough $n$ (see Figure 1), makes our bound tighter than others.

Adding a regularization term in the objective (8) is important as it stabilizes $\hat{h}(Z_{<m})$ and $\hat{h}(Z_{\geq m})$; a similar effect is achieved with methods like gradient descent as they essentially have a "built-in" regularization. For very small sample sizes, the regularization in (8) may not be enough to ensure that $\hat{h}(Z_{<m})$ and $\hat{h}(Z_{\geq m})$ are close to $\hat{h}(Z_{\leq n})$, in which case $V_n$ need not be necessarily small. In particular, this is the case for the Haberman and the breast cancer datasets where the advantage of our bound is not fully leveraged, and Maurer's bound is smaller.

## 5  Theoretical Motivation of the Bound

In this section, we study the behavior of our bound (5) under a Bernstein condition:

**Definition 6. [Bernstein Condition (BC)]** *The learning problem $(\mathbf{D}, \ell, \mathcal{H})$ satisfies the $(\beta, B)$-Bernstein condition, for $\beta \in [0, 1]$ and $B > 0$, if for all $h \in \mathcal{H}$,*

$$\mathbb{E}_{Z \sim \mathbf{D}}\left[\left(\ell_h(Z) - \ell_{h_*}(Z)\right)^2\right] \leq B \cdot \mathbb{E}_{Z \sim \mathbf{D}}\left[\ell_h(Z) - \ell_{h_*}(Z)\right]^\beta,$$

*where $h_* \in \arg\inf_{h \in \mathcal{H}} \mathbb{E}_{Z \sim \mathbf{D}}\left[\ell_h(Z)\right]$ is a risk minimizer within the closer of $\mathcal{H}$.*

The Bernstein condition [3, 5, 6, 15, 22] essentially characterizes the "easiness" of the learning problem; it implies that the variance in the excess loss random variable $\ell_h(Z) - \ell_{h_*}(Z)$ gets smaller the closer the risk of hypothesis $h \in \mathcal{H}$ gets to that of the risk minimizer $h_*$. For bounded loss functions, the BC with $\beta = 0$ always holds. The BC with $\beta = 1$ (the "easiest" learning setting) is also known as the *Massart noise condition* [25]; it holds in our experiment with synthetic data in Section 4, and also, *e.g.*, whenever $\mathcal{H}$ is convex and $h \mapsto \ell_h(z)$ is exp-concave, for all $z \in \mathcal{Z}$ [15, 31]. For more examples of learning settings where a BC holds see [22, Section 3].

Our aim in this section is to give an upper-bound on the infimum term involving $V_n$ in (5), under a BC, in terms of the complexity $\mathrm{COMP}_n$ and the excess risks $\bar{L}(P_n)$, $\bar{L}(Q(Z_{>m}))$, and $\bar{L}(Q(Z_{\leq m}))$,

where for a distribution $P \in \mathcal{P}(\mathcal{H})$, the excess risk is defined by

$$\bar{L}(P) \coloneqq \mathbb{E}_{h \sim P}\left[\mathbb{E}_{Z \sim \mathbf{D}}\left[\ell_h(Z)\right]\right] - \mathbb{E}_{Z \sim \mathbf{D}}\left[\ell_{h_*}(Z)\right].$$

In the next theorem, we denote $Q_{\leq m} \coloneqq Q(Z_{\leq m})$ and $Q_{>m} \coloneqq Q(Z_{>m})$, for $m \in [n]$. To simplify the presentation further (and for consistency with Section 4), we assume that $Q$ is chosen such that

$$Q(Z_{>i}) = Q_{>m}, \text{ for } 1 \leq i \leq m, \quad \text{and} \quad Q(Z_{<j}) = Q_{\leq m}, \text{ for } m < j \leq n. \tag{9}$$

**Theorem 7.** *Let $\mathcal{G}$ and $\pi$ be as in (7), $\delta \in ]0,1[$, and $\varepsilon_{\delta,n} = 2\ln\frac{1}{\delta \cdot \pi(\eta)} = 2\ln\frac{|\mathcal{G}|}{\delta}$, $\eta \in \mathcal{G}$. If the $(\beta, B)$-Bernstein condition holds with $\beta \in [0,1]$ and $B > 0$, then for any learning algorithms $P$ and $Q$ (with $Q$ satisfying (9)), there exists a $\mathcal{C} > 0$, such that $\forall n \geq 1$ and $m = n/2$, with probability at least $1 - \delta$,*

$$\frac{1}{\mathcal{C}} \cdot \inf_{\eta \in \mathcal{G}} \left\{ c_\eta \cdot V_n + \frac{\mathrm{COMP}_n + \varepsilon_{\delta,n}}{\eta \cdot n} \right\} \leq \bar{L}(P_n) + \bar{L}(Q_{\leq m}) + \bar{L}(Q_{>m})$$

$$+ \left( \frac{\mathrm{COMP}_n + \varepsilon_{\delta,n}}{n} \right)^{\frac{1}{2-\beta}} + \frac{\mathrm{COMP}_n + \varepsilon_{\delta,n}}{n}. \tag{10}$$

In addition to the "ESI" tools provided in Section 6 and Lemma 13, the proof of Theorem 7, presented in Appendix E, also uses an "ESI version" of the Bernstein condition due to [22].

First note that the only terms in our main bound (5), other than the infimum on the LHS of (10), are the empirical error $L_n(P_n)$ and a $\tilde{O}(1/\sqrt{n})$-complexity-free term which is typically smaller than $\sqrt{\mathrm{KL}(P_n\|P_0)/n}$ (*e.g.* when the dimension of $\mathcal{H}$ is large enough). The term $\sqrt{\mathrm{KL}(P_n\|P_0)/n}$ is often the dominating one in other PAC-Bayesian bounds when $\liminf_{n\to\infty} L_n(P_n) > 0$.

Now consider the remaining term in our main bound, which matches the infimum term on the LHS of (10), and let us choose algorithm $P$ as per Remark 5, so that $\mathrm{COMP}_n = 2\mathrm{KL}(P_n\|P_0)$. Suppose that, with high probability (w.h.p.), $\mathrm{KL}(P_n\|P_0)/n$ converges to 0 for $n \to \infty$ (otherwise no PAC-Bayesian bound would converge to 0), then $(\mathrm{COMP}_n/n)^{1/(2-\beta)} + \mathrm{COMP}_n/n$—essentially the sum of the last two terms on the RHS of (10)—converges to 0 at a faster rate than $\sqrt{\mathrm{KL}(P_n\|P_0)/n}$ w.h.p. for $\beta > 0$, and at equal rate for $\beta = 0$. Thus, in light of Theorem 7, to argue that our bound can be better than others (still when $\liminf_{n\to\infty} L_n(P_n) > 0$), it remains to show that there exist algorithms $P$ and $Q$ for which the sum of the excess risks on the RHS of (10) is smaller than $\sqrt{\mathrm{KL}(P_n\|P_0)/n}$.

One choice of estimator with small excess risk is the *Empirical Risk Minimizer* (ERM). When $m = n/2$, if one chooses $Q$ such that it outputs a Dirac around the ERM on a given sample, then under a BC with exponent $\beta$ and for "parametric" $\mathcal{H}$ (such as the $d$-dimensional linear classifiers in Sec. 4), $\bar{L}(Q_{\leq m})$ and $\bar{L}(Q_{>m})$ are of order $\tilde{O}\left(n^{-1/(2-\beta)}\right)$ w.h.p. [3, 19]. However, setting $P_n \equiv \delta(\mathrm{ERM}(Z_{\leq n}))$ is not allowed, since otherwise $\mathrm{KL}(P_n\|P_0) = \infty$. Instead one can choose $P_n$ to be the generalized-Bayes/Gibbs posterior. In this case too, under a BC with exponent $\beta$ and for parametric $\mathcal{H}$, the excess risk is of order $\tilde{O}\left(n^{-1/(2-\beta)}\right)$ w.h.p. for clever choices of prior $P_0$ [3, 19].

## 6 Detailed Analysis

We start this section by presenting the convenient ESI notation and use it to present our main technical Lemma 13 (proofs of the ESI results are in Appendix D). We then continue with a proof of Theorem 3.

**Definition 8. [ESI (*Exponential Stochastic Inequality, pronounce as:easy*) 19, 22]** *Let $\eta > 0$, and $X, Y$ be any two random variables with joint distribution $\mathbf{D}$. We define*

$$X \trianglelefteq_\eta^{\mathbf{D}} Y \iff X - Y \trianglelefteq_\eta^{\mathbf{D}} 0 \iff \mathbb{E}_{(X,Y)\sim\mathbf{D}}\left[e^{\eta(X-Y)}\right] \leq 1. \tag{11}$$

Definition 8 can be extended to the case where $\eta = \hat{\eta}$ is also a random variable, in which case the expectation in (11) needs to be replaced by the expectation over the joint distribution of $(X, Y, \hat{\eta})$. When no ambiguity can arise, we omit $\mathbf{D}$ from the ESI notation. Besides simplifying notation, ESIs are useful in that they simultaneously capture "with high probability" and "in expectation" results:

**Proposition 9. [ESI Implications]** *For fixed $\eta > 0$, if $X \trianglelefteq_\eta Y$ then $\mathbb{E}[X] \leq \mathbb{E}[Y]$. For both fixed and random $\hat{\eta}$, if $X \trianglelefteq_{\hat{\eta}} Y$, then $\forall \delta \in ]0,1[$, $X \leq Y + \frac{\ln\frac{1}{\delta}}{\hat{\eta}}$, with probability at least $1 - \delta$.*

In the next proposition, we present two results concerning transitivity and additive properties of ESI:

**Proposition 10. [ESI Transitivity and Chain Rule]** *(a) Let $Z_1, \ldots, Z_n$ be any random variables on $\mathcal{Z}$ (not necessarily independent). If for some $(\gamma_i)_{i \in [n]} \in ]0, +\infty[^n$, $Z_i \trianglelefteq_{\gamma_i} 0$, for all $i \in [n]$, then*

$$\sum_{i=1}^{n} Z_i \trianglelefteq_{\nu_n} 0, \quad \text{where } \nu_n := \left( \sum_{i=1}^{n} \frac{1}{\gamma_i} \right)^{-1} \text{ (so if } \forall i \in [n], \gamma_i = \gamma > 0 \text{ then } \nu_n = \gamma/n).$$

*(b) Suppose now that $Z_1, \ldots, Z_n$ are i.i.d. and let $X : \mathcal{Z} \times \bigcup_{i=1}^{n} \mathcal{Z}^i \to \mathbb{R}$ be any real-valued function. If for some $\eta > 0$, $X(Z_i; z_{<i}) \trianglelefteq_\eta 0$, for all $i \in [n]$ and all $z_{<i} \in \mathcal{Z}^{i-1}$, then $\sum_{i=1}^{n} X(Z_i; Z_{<i}) \trianglelefteq_\eta 0$.*

We now give a basic PAC-Bayesian result for the ESI context:

**Proposition 11. [ESI PAC-Bayes]** *Fix $\eta > 0$ and let $\{Y_h : h \in \mathcal{H}\}$ be any family of random variables such that for all $h \in \mathcal{H}$, $Y_h \trianglelefteq_\eta 0$. Let $P_0$ be any distribution on $\mathcal{H}$ and let $P : \bigcup_{i=1}^{n} \mathcal{Z}^i \to \mathcal{P}(\mathcal{H})$ be a learning algorithm. We have:*

$$\mathbb{E}_{h \sim P_n}[Y_h] \trianglelefteq_\eta \frac{\text{KL}(P_n \| P_0)}{\eta}, \quad \text{where } P_n := P(Z_{\leq n}).$$

In many applications (especially for our main result) it is desirable to work with a random (*i.e.* data-dependent) $\eta$ in the ESI inequalities; one can tune $\eta$ after seeing the data.

**Proposition 12. [ESI from fixed to random $\eta$]** *Let $\mathcal{G}$ be a countable subset of $]0, +\infty[$ and let $\pi$ be a prior distribution over $\mathcal{G}$. Given a countable collection $\{Y_\eta : \eta \in \mathcal{G}\}$ of random variables satisfying $Y_\eta \trianglelefteq_\eta 0$, for all fixed $\eta \in \mathcal{G}$, we have, for arbitrary estimator $\hat{\eta}$ with support on $\mathcal{G}$,*

$$Y_{\hat{\eta}} \trianglelefteq_{\hat{\eta}} \frac{-\ln \pi(\hat{\eta})}{\hat{\eta}}.$$

The following key lemma, which is of independent interest, is central to our main result.

**Lemma 13. [Key result: *un*-expected Bernstein]** *Let $X \sim \mathbf{D}$ be a random variable bounded from above by $b > 0$ almost surely, and let $\vartheta(u) := (-\ln(1-u) - u)/u^2$. For all $0 < \eta < 1/b$, we have (a):*

$$\mathbb{E}[X] - X \trianglelefteq_\eta^{\mathbf{D}} c \cdot X^2, \quad \text{for all } c \geq \eta \cdot \vartheta(\eta b). \tag{12}$$

*(b): The result is tight; for every $c < \eta \cdot \vartheta(\eta b)$, there exists a distribution $\mathbf{D}$ so that (12) does not hold.*

Lemma 13 is reminiscent of the following slight variation of Bernstein's inequality [12]; let $X$ be any random variable bounded from *below* by $-b$, and let $\kappa(x) := (e^x - x - 1)/x^2$. For all $\eta > 0$, we have

$$\mathbb{E}[X] - X \trianglelefteq_\eta s \cdot \mathbb{E}[X^2], \quad \text{for all } s \geq \eta \cdot \kappa(\eta b). \tag{13}$$

Note that the un-expected Bernstein Lemma 13 has the $X^2$ lifted out of the expectation. In Appendix G, we prove (13) and compare it to standard versions of Bernstein. We also compare (12) to the related but distinct empirical Bernstein inequality due to [27, Theorem 4]. We now prove part (a) of Lemma 13, which follows easily from the proof of an existing result [16, 21]. Part (b) is novel; its proof is postponed to Appendix F.

**Proof of Lemma 13-Part (a).** [16] (see also [21]) showed in the proof of their lemma 4.1 that

$$\exp(\lambda \xi - \lambda^2 \vartheta(\lambda) \xi^2) \leq 1 + \lambda \xi, \quad \text{for all } \lambda \in [0, 1[ \text{ and } \xi \geq -1. \tag{14}$$

Letting $\eta = \lambda/b$ and $\xi = -X/b$, (14) becomes,

$$\exp(-\eta X - \eta^2 \vartheta(\eta b) X^2) \leq 1 - \eta X, \quad \text{for all } \eta \in ]0, 1/b[. \tag{15}$$

Taking expectation on both sides of (15) and using the fact that $1 - \eta \mathbb{E}[X] \leq \exp(-\eta \mathbb{E}[X])$ on the RHS of the resulting inequality, leads to (12). □

**Proof of Theorem 3.** Let $\eta \in ]0, 1/b[$ and $c_\eta := \eta \cdot \vartheta(\eta b)$. For $1 \leq i \leq m < j \leq n$, define

$$X_h(Z_i; z_{>i}) := \ell_h(Z_i) - \mathbb{E}_{h' \sim Q(z_{>i})}[\ell_{h'}(Z_i)], \quad \text{for } z_{>i} \in \mathcal{Z}^{n-i},$$

$$\tilde{X}_h(Z_j; z_{<j}) := \ell_h(Z_j) - \mathbb{E}_{h' \sim Q(z_{<j})}[\ell_{h'}(Z_j)], \quad \text{for } z_{<j} \in \mathcal{Z}^{j-1}.$$

Since $\ell$ is bounded from above by $b$, Lemma 13 implies that for all $h \in \mathcal{H}$ and $1 \le i \le m < j \le n$,

$$\forall z_{>i} \in \mathcal{Z}^{n-i}, \quad Y_h^\eta(Z_i; z_{>i}) := \mathbb{E}_{Z_i' \sim \mathbf{D}}\left[X_h(Z_i'; z_{>i})\right] - X_h(Z_i; z_{>i}) - c_\eta \cdot X_h(Z_i; z_{>i})^2 \trianglelefteq_\eta 0,$$

$$\forall z_{<j} \in \mathcal{Z}^{j-1}, \quad \tilde{Y}_h^\eta(Z_j; z_{<j}) := \mathbb{F}_{Z_j' \sim \mathbf{D}}\left[\tilde{X}_h(Z_j'; z_{<j})\right] - \tilde{X}_h(Z_j; z_{<j}) - c_\eta \cdot \tilde{X}_h(Z_j; z_{<j})^2 \trianglelefteq_\eta 0,$$

Since $Z_1, \ldots, Z_n$ are i.i.d. we can chain the ESIs above using Proposition 10-(b) to get:

$$S := \sum_{i=1}^m Y_h^\eta(Z_i; z_{>i}) \trianglelefteq_\eta 0, \quad \tilde{S} := \sum_{j=m+1}^n \tilde{Y}_h^\eta(Z_j; z_{<j}) \trianglelefteq_\eta 0. \tag{16}$$

Applying PAC-Bayes (Proposition 11) to $S$ and $\tilde{S}$ in (16) with priors $P(Z_{>m})$ and $P(Z_{\le m})$, respectively, and common posterior $P_n = P(Z_{\le n})$ on $\mathcal{H}$, we get, with $\mathrm{KL}_{>m} := \mathrm{KL}(P_n \| P(Z_{>m}))$ and $\mathrm{KL}_{\le m} := \mathrm{KL}(P_n \| P(Z_{\le m}))$:

$$\mathbb{E}_{h \sim P_n}\left[\sum_{i=1}^m Y_h^\eta(Z_i; z_{>i})\right] - \frac{\mathrm{KL}_{>m}}{\eta} \trianglelefteq_\eta 0, \quad \mathbb{E}_{h \sim P_n}\left[\sum_{j=m+1}^n \tilde{Y}_h^\eta(Z_j; z_{<j})\right] - \frac{\mathrm{KL}_{\le m}}{\eta} \trianglelefteq_\eta 0.$$

We now apply Proposition 10-(a) to chain these two ESIs, which yields

$$\mathbb{E}_{h \sim P_n}\left[\sum_{i=1}^m Y_h^\eta(Z_i; z_{>i}) + \sum_{j=m+1}^n \tilde{Y}_h^\eta(Z_j; Z_{<j})\right] \trianglelefteq_{\frac{\eta}{2}} \frac{\mathrm{KL}(P_n \| P(Z_{>m})) + \mathrm{KL}(P_n \| P(Z_{\le m}))}{\eta}.$$

With the prior $\pi$ on $\mathcal{G}$, we have for any $\hat{\eta} = \hat{\eta}(Z_{\le n}) \in \mathcal{G} \subset [1/\sqrt{nb^2}, 1/b[$ (see Proposition 12),

$$\mathbb{E}_{h \sim P_n}\left[\sum_{i=1}^m Y_h^{\hat{\eta}}(Z_i; z_{>i}) + \sum_{j=m+1}^n \tilde{Y}_h^{\hat{\eta}}(Z_j; Z_{<j})\right] \trianglelefteq_{\frac{\hat{\eta}}{2}} \frac{\mathrm{COMP}_n}{\hat{\eta}} - \frac{2\ln \pi(\hat{\eta})}{\hat{\eta}}, \quad i.e.,$$

$$n \cdot (L(P_n) - L_n(P_n)) \quad \trianglelefteq_{\frac{\hat{\eta}}{2}} \quad n \cdot c_{\hat{\eta}} \cdot V_n + \frac{\mathrm{COMP}_n + 2\ln \frac{1}{\pi(\hat{\eta})}}{\hat{\eta}} \quad +$$

$$\left[\sum_{i=1}^m \left(\mathbb{E}_{Z_i' \sim \mathbf{D}}\left[\bar{\ell}_{Q_{>i}}(Z_i')\right] - \bar{\ell}_{Q_{>i}}(Z_i)\right) + \sum_{j=m+1}^n \left(\mathbb{E}_{Z_j' \sim \mathbf{D}}\left[\bar{\ell}_{Q_{<j}}(Z_j')\right] - \bar{\ell}_{Q_{<j}}(Z_j)\right)\right], \tag{17}$$

where $\bar{\ell}_{Q_{>i}}(Z_i) := \mathbb{E}_{h \sim Q(Z_{>i})}\left[\ell_h(Z_i)\right]$ and $\bar{\ell}_{Q_{<j}}(Z_j) := \mathbb{E}_{h \sim Q(Z_{<j})}\left[\ell_h(Z_j)\right]$. Let $U_n$ denote the quantity between the square brackets in (17). Using the un-expected Bernstein Lemma 13, together with Proposition 12, we get for any estimator $\hat{\nu}$ on $\mathcal{G}$:

$$U_n \trianglelefteq_{\hat{\nu}} c_{\hat{\nu}} \cdot \left(\sum_{i=1}^m \mathbb{E}_{h' \sim Q(Z_{>i})}\left[\ell_{h'}(Z_i)^2\right] + \sum_{j=m+1}^n \mathbb{E}_{h' \sim Q(Z_{<j})}\left[\ell_{h'}(Z_j)^2\right]\right) + \frac{\ln \frac{1}{\pi(\hat{\nu})}}{\hat{\nu}}. \tag{18}$$

By chaining (18) and (17) using Proposition 10-(a) and dividing by $n$, we get:

$$L(P_n) \trianglelefteq_{\frac{n\hat{\eta}\hat{\nu}}{\hat{\eta}+2\hat{\nu}}} L_n(P_n) + c_{\hat{\eta}} \cdot V_n + \frac{\mathrm{COMP}_n + 2\ln \frac{1}{\pi(\hat{\eta})}}{\hat{\eta} \cdot n} + c_{\hat{\nu}} \cdot V_n' + \frac{\ln \frac{1}{\pi(\hat{\nu})}}{\hat{\nu} \cdot n}. \tag{19}$$

We now apply Proposition 9 to (19) to obtain the following inequality with probability at least $1 - \delta$:

$$L(P_n) \le L_n(P_n) + \left[c_{\hat{\eta}} \cdot V_n + \frac{\mathrm{COMP}_n + 2\ln \frac{1}{\pi(\hat{\eta}) \cdot \delta}}{\hat{\eta} \cdot n}\right] + \left\{c_{\hat{\nu}} \cdot V_n' + \frac{\ln \frac{1}{\pi(\hat{\nu}) \cdot \delta}}{\hat{\nu} \cdot n}\right\}. \tag{20}$$

Inequality (5) follows after picking $\hat{\nu}$ and $\hat{\eta}$ to be, respectively, estimators which achieve the infimum over the closer of $\mathcal{G}$ of the quantities between braces and square brackets in (20). $\square$

# 7 Conclusion and Future Work

The main goal of this paper was to introduce a new PAC-Bayesian bound based on a new proof technique; we also theoretically motivated the bound in terms of a Bernstein condition. The simple experiments we provided are to be considered as a basic sanity check—in future work, we plan to put the bound to real practical use by applying it to deep nets in the style of, *e.g.*, [42].

**Acknowledgments**

An anonymous referee made some highly informed remarks on our paper, which led us to substantially rewrite the paper and made us understand our own work much better. Part of this work was performed while Zakaria Mhammedi was interning at the Centrum Wiskunde & Informatica (CWI). This work was also supported by the Australian Research Council and Data61.

## Footnotes

[1]An alternative form of empirical Bernstein inequality appears in [41], based on an inequality due to [11].

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
