[Supplementary Material · main.pdf]

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

## A  Informed Priors

Any bound of the form of (1) with $\text{COMP}_n = \text{KL}(P_n \| P_0)$ can be applied in a way as to replace this KL term by $\text{KL}(P_n \| P(Z_{>m})) + \text{KL}(P_n \| P(Z_{\leq m}))$, and thus making use of "informed priors". For this, it suffices to apply the bound on each part of the sample, *i.e.* $Z_{>m}$ and $Z_{\leq m}$, and then combine the resulting bounds with a union bound. In fact, suppose that (1) holds with $R_n = L_n(P_n)$ and $\mathcal{C} = 0$, and let $\delta \in\, ]0, 1[$. Applying the bound on the second part of the sample $Z_{>m}$ with prior $P(Z_{\leq m})$ and posterior $P_n$, we get, with probability at least $1 - \delta$,

$$
\begin{aligned}
L(P_n) - L_{>m}(P_n) \leq \mathcal{P} \cdot &\sqrt{\frac{L_{>m}(P_n) \cdot (\text{KL}(P_n \| P(Z_{\leq m})) + \varepsilon_{\delta, n-m})}{n-m}} \\
&+ \mathcal{A} \cdot \frac{\text{KL}(P_n \| P(Z_{\leq m})) + \varepsilon_{\delta, n-m}}{n-m},
\end{aligned}
\tag{25}
$$

where $L_{>m}(P_n) \coloneqq \frac{1}{n-m} \sum_{j=m+1}^{n} \mathbb{E}_{h \sim P_n}[\ell_h(Z_j)]$. Similarly, applying the bound on the first half of the sample $Z_{\leq m}$ with prior $P(Z_{>m})$ and posterior $P_n$, we get, with probability at least $1 - \delta$,

$$
\begin{aligned}
L(P_n) - L_{\leq m}(P_n) \leq \mathcal{P} \cdot &\sqrt{\frac{L_{\leq m}(P_n) \cdot (\text{KL}(P_n \| P(Z_{>m})) + \varepsilon_{\delta, m})}{m}} \\
&+ \mathcal{A} \cdot \frac{\text{KL}(P_n \| P(Z_{>m})) + \varepsilon_{\delta, m}}{m},
\end{aligned}
\tag{26}
$$

where $L_{\leq m}(P_n) \coloneqq \frac{1}{m} \sum_{i=1}^{m} \mathbb{E}_{h \sim P_n}[\ell_h(Z_i)]$. Let $p \coloneqq m/n$ and $q \coloneqq (n-m)/n$ (note that $p + q = 1$). Applying a union bound and adding $q \times$ (25) with $p \times$ (26), yields the bound

$$
\begin{aligned}
L(P_n) - L_n(P_n) \leq \mathcal{P} \cdot &\sqrt{\frac{2 L_n(P_n) \cdot (\text{KL}(P_n \| P(Z_{>m})) + \text{KL}(P_n \| P(Z_{\leq m})) + \bar{\varepsilon}_{\delta, n})}{n}} \\
&+ \mathcal{A} \cdot \frac{\text{KL}(P_n \| P(Z_{>m})) + \text{KL}(P_n \| P(Z_{\leq m})) + \bar{\varepsilon}_{\delta, n}}{n},
\end{aligned}
\tag{27}
$$

with probability at least $1 - \delta$, where $\bar{\varepsilon}_{\delta, n} \coloneqq \varepsilon_{\delta/2, m} + \varepsilon_{\delta/2, n-m}$. To get to (27), we also used the fact that $\sqrt{x} + \sqrt{y} \leq \sqrt{2(x+y)}$, for all $x, y \in \mathbb{R}_{\geq 0}$.

The above trick does not directly apply to Maurer's bound in (2) (since the dependence on $L(P_n)$ is not linear). Instead, one can use the joint convexity of the binary Kullback-Leibler divergence $\text{kl}$ in its two arguments as in the following proof of Lemma 4:

**Proof of Lemma 4.** Let $\delta \in\, ]0, 1[$. We can write $L_n(P_n)$ as

$$
L_n(P_n) = \frac{p}{m} \sum_{i=1}^{m} \mathbb{E}_{h \sim P_n}[\ell_h(Z_i)] + \frac{q}{n-m} \sum_{j=m+1}^{n} \mathbb{E}_{h \sim P_n}[\ell_h(Z_j)],
$$

where $p \coloneqq m/n$ and $q \coloneqq (n-m)/n$ (note that $p + q = 1$). Let us denote

$$
L_{\leq m}(P_n) \coloneqq \frac{1}{m} \sum_{i=1}^{m} \mathbb{E}_{h \sim P_n}[\ell_h(Z_i)] \ \text{ and } \ L_{>m}(P_n) \coloneqq \frac{1}{n-m} \sum_{j=m+1}^{n} \mathbb{E}_{h \sim P_n}[\ell_h(Z_j)].
$$

By the joint convexity of the binary Kullback-Leibler divergence $\text{kl}$ in its two arguments, we have

$$
\begin{aligned}
\text{kl}(L(P_n) \| L_n(P_n)) &= \text{kl}(p L(P_n) + q L(P_n) \| p L_{\leq m}(P_n) + q L_{>m}(P_n)), \\
&\leq p \cdot \text{kl}(L(P_n) \| L_{\leq m}(P_n)) + q \cdot \text{kl}(L(P_n) \| L_{>m}(P_n)), \\
&\leq p \cdot \frac{\text{KL}(P_n \| P(Z_{>m})) + \ln \frac{4\sqrt{m}}{\delta}}{m}, \\
&\quad + q \cdot \frac{\text{KL}(P_n \| P(Z_{\leq m})) + \ln \frac{4\sqrt{n-m}}{\delta}}{n-m},
\end{aligned}
\tag{28}
$$

with probability at least $1 - \delta$, where the last inequality follows by Maurer's bound (2) and the union bound. Substituting the expressions of $p$ and $q$ in (28) yields the desired result. $\qquad\square$

# B Biasing

A PAC-Bayes bound similar to the one in our Corollary 1 can be obtained from the TS bound. For this, the TS bound must be applied twice, once on each part of the sample (*i.e.* $Z_{\leq m}$ and $Z_{>m}$) to *biased* losses. We demonstrate this in what follows.

Let $\hat{h} : \bigcup_{i=1}^{n} \mathcal{Z}^i \to \mathcal{H}$ be any estimator. The TS bound can be expressed in the form of (1) with $\text{COMP}_n = \text{KL}(P_n \| P_0)$, $\mathcal{C} = 0$, and $R_n = \mathbb{E}_{h \sim P_n}[\text{Var}_n[\ell_h(Z)]]$, where $\text{Var}_n[X]$ denotes the empirical variance. Applying the TS bound on the second part of the sample $Z_{>m}$ with prior $P_0$ and posterior $P_n$, and with the biased loss $\tilde{\ell}_h(Z) = \ell_h(Z) - \ell_{\hat{h}(Z_{\leq m})}(Z)$, gives

$$\tilde{L}(P_n) - \tilde{L}_{>m}(P_n) \leq \mathcal{P} \cdot \sqrt{\frac{\mathbb{E}_{h \sim P_n}[\text{Var}_{>m}[\tilde{\ell}_h(Z)]] \cdot (\text{KL}(P_n \| P_0) + \varepsilon_{\delta, n-m})}{n-m}} + \mathcal{A} \cdot \frac{\text{KL}(P_n \| P_0) + \varepsilon_{\delta, n-m}}{n-m}, \tag{29}$$

with probability at least $1 - \delta$, where $\text{Var}_{>m}[X] \coloneqq \frac{1}{n-m} \sum_{i=m+1}^{n} \left( X_i - \frac{1}{n-m} \sum_{j=m+1}^{n} X_j \right)^2$, $\tilde{L}(P_n) \coloneqq \mathbb{E}_{h \sim P_n}[\mathbb{E}_{Z \sim \mathbf{D}}[\tilde{\ell}_h(Z)]]$, and $\tilde{L}_{>m}(P_n) \coloneqq \frac{1}{n-m} \sum_{j=m+1}^{n} \mathbb{E}_{h \sim P_n}[\tilde{\ell}_h(Z_j)]$.

Doing the same on the first part of the sample $Z_{\leq m}$, but now with the loss $\check{\ell}_h(Z) \coloneqq \ell_h(Z) - \ell_{\hat{h}(Z_{>m})}(Z)$, yields

$$\check{L}(P_n) - \check{L}_{\leq m}(P_n) \leq \mathcal{P} \cdot \sqrt{\frac{\mathbb{E}_{h \sim P_n}[\text{Var}_{\leq m}[\check{\ell}_h(Z)]] \cdot (\text{KL}(P_n \| P_0) + \varepsilon_{\delta, m})}{m}} + \mathcal{A} \cdot \frac{\text{KL}(P_n \| P_0) + \varepsilon_{\delta, m}}{m}, \tag{30}$$

with probability at least $1 - \delta$, where $\text{Var}_{\leq m}[X] \coloneqq \frac{1}{m} \sum_{i=1}^{m} \left( X_i - \frac{1}{m} \sum_{j=1}^{m} X_j \right)^2$, $\check{L}(P_n) \coloneqq \mathbb{E}_{h \sim P_n}[\mathbb{E}_{Z \sim \mathbf{D}}[\check{\ell}_h(Z)]]$, and $\check{L}_{\leq m}(P_n) \coloneqq \frac{1}{m} \sum_{i=1}^{m} \mathbb{E}_{h \sim P_n}[\check{\ell}_h(Z_i)]$.

Two more applications of the TS bound with prior and posterior equal to $P_0$, yields,

$$L(\hat{h}(Z_{\leq m})) - L_{>m}(\hat{h}(Z_{\leq m})) \leq \mathcal{P} \cdot \sqrt{\frac{\text{Var}_{>m}[\ell_{\hat{h}(Z_{\leq m})}(Z) \cdot \varepsilon_{\delta/2, n-m}}{n-m}} + \frac{\mathcal{A} \cdot \varepsilon_{\delta/2, n-m}}{n-m}, \text{ and} \tag{31}$$

$$L(\hat{h}(Z_{>m})) - L_{\leq m}(\hat{h}(Z_{>m})) \leq \mathcal{P} \cdot \sqrt{\frac{\text{Var}_{\leq m}[\ell_{\hat{h}(Z_{>m})}(Z) \cdot \varepsilon_{\delta/2, m}}{m}} + \frac{\mathcal{A} \cdot \varepsilon_{\delta/2, m}}{m}, \tag{32}$$

with probability at least $1 - \delta$, where

$$L_{\leq m}(\hat{h}(Z_{>m})) \coloneqq \frac{1}{m} \sum_{i=1}^{m} \ell_{\hat{h}(Z_{>m})}(Z_i) \quad \text{and} \quad L_{>m}(\hat{h}(Z_{\leq m})) \coloneqq \frac{1}{n-m} \sum_{j=m+1}^{n} \ell_{\hat{h}(Z_{\leq m})}(Z_j).$$

Let $p = m/n$ and $q = (n-m)/n$. Applying a union bound and combining (29)-(32), as

$$q \times ((29) + (31)) + p \times ((30) + (32)),$$

yields a bound of the form (1) with

$$R'_n = p \cdot \text{Var}_{\leq m}[\ell_{\hat{h}(Z_{>m})}(Z)] + q \cdot \text{Var}_{<m}[\ell_{\hat{h}(Z_{\leq m})}(Z)] \leq V'_n,$$

$$R_n = p \cdot \mathbb{E}_{h \sim P_n}[\text{Var}_{\leq m}[\check{\ell}_h(Z)]] + q \cdot \mathbb{E}_{h \sim P_n}[\text{Var}_{>m}[\tilde{\ell}_h(Z)]] \leq V_n,$$

where $V'_n$ and $V_n$ are as in Corollary 1.

**A Direct Approach.** Though the steps above lead to a bound similar to ours in Corollary 1, the constants involved may not be optimal. We now re-derive a modification of the TS bound with a $V_n$ term like in Corollary 1, and with tighter constants. The proof techniques used here are the same as those used in the proof of Theorem 3. For $\eta \in ]0, 1/b[$ (where $b > 0$ is an upper-bound on the loss $\ell$) and $m \in [2..n]$, define

$$s_\eta \coloneqq \eta \cdot \kappa(\eta b), \quad \text{where} \quad \kappa(\eta) \coloneqq (e^\eta - \eta - 1)/\eta^2,$$

$$\text{and} \quad \tilde{c}_\eta \coloneqq \frac{s_\eta m}{2m-2} \left( 1 + \frac{\eta m}{2m-2} \right)^{-1}, \quad \lambda(\eta) \coloneqq \frac{\eta \beta(\eta)}{\eta + \beta(\eta)}, \quad \text{where} \quad \beta(\eta) \coloneqq \eta + \frac{\eta^2 m^2}{2m-2}.$$

We assume that $n > 2$ is even in the next theorem. We remind the reader of the definitions

$$\text{Var}_{\leq m}[X] := \frac{1}{m} \sum_{i=1}^{m} \left( X_i - \frac{1}{m} \sum_{j=1}^{m} X_j \right)^2 \text{ and } \text{Var}_{>m}[X] := \frac{1}{n-m} \sum_{i=m+1}^{n} \left( X_i - \frac{1}{n-m} \sum_{j=m+1}^{n} X_j \right)^2.$$

**Theorem 14.** **[New PAC-Bayes Empirical Bernstein Bound]** *Let $Z_1, \ldots, Z_n$ be i.i.d. with $Z_1 \sim \mathbf{D}$. Let $m = n/2 > 1$ and $\pi$ be any distribution with support on a finite or countable grid $\mathcal{G} \subset ]0, 1/b[$. For any $\delta \in ]0, 1[$, learning algorithm $P : \bigcup_{i=1}^{n} \mathcal{Z}^i \to \mathcal{P}(\mathcal{H})$, and estimator $\hat{h} : \bigcup_{i=1}^{n} \mathcal{Z}^i \to \mathcal{H}$, we have,*

$$L(P_n) \leq L_n(P_n) + \inf_{\eta \in \mathcal{G}} \left\{ \tilde{c}_\eta \cdot G_n + \frac{\text{COMP}_n + 2 \ln \frac{1}{\delta \cdot \pi(\eta)}}{\lambda(\eta) \cdot n} \right\} + \inf_{\nu \in \mathcal{G}} \left\{ \tilde{c}_\nu \cdot G_n' + \frac{\ln \frac{1}{\delta \cdot \pi(\nu)}}{\lambda(\nu) \cdot n} \right\},$$

*with probability at least $1 - \delta$, where $\text{COMP}_n$, $G_n'$, and $G_n$ are the random variables defined by:*

$$\text{COMP}_n := \text{KL}(P_n \| P(Z_{\leq m})) + \text{KL}(P_n \| P(Z_{>m})),$$

$$G_n' := \text{Var}_{>m}\left[ \ell_{\hat{h}(Z_{\leq m})}(Z) \right] + \text{Var}_{\leq m}\left[ \ell_{\hat{h}(Z_{>m})}(Z) \right],$$

$$G_n := \mathbb{E}_{h \sim P_n}\left[ \text{Var}_{>m}\left[ \ell_h(Z) - \ell_{\hat{h}(Z_{\leq m})}(Z) \right] + \text{Var}_{\leq m}\left[ \ell_h(Z) - \ell_{\hat{h}(Z_{>m})}(Z) \right] \right].$$

Note that since $\text{Var}_{\leq m}(X) \leq \sum_{i=1}^{m} X_i^2/m$ and $\text{Var}_{>m}(X) \leq \sum_{i=m+1}^{n} X_i^2/m$, we have

$$G_n \leq V_n \quad \text{and} \quad G_n' \leq V_n',$$

where $V_n$ and $V_n'$ are defined in (4) and (3), respectively. However, one cannot directly compare $G_n$ to the $V_n$ defined in Theorem 3, since the latter uses "online" posteriors ($Q(Z_{>i})$) and $Q(Z_{<j})$ which get closer and closer to the posterior $Q(Z_{\leq n})$ based on the full sample.

To prove Theorem 14, we need the following self-bounding property of the empirical variance [27]:

$$m \, \text{Var}[X] \trianglelefteq_\eta \frac{m^2}{m-1} \text{Var}_m[X] - \frac{\eta m^2}{2m-2} \text{Var}[X], \tag{33}$$

for any $\eta > 0$ and any bounded random variable $X$, where $\text{Var}_m[X]$ is either $\text{Var}_{>m}[X]$ or $\text{Var}_{\leq m}[X]$ (recall that $m = n/2$). Re-arranging (33) and dividing by $(1 + \eta m/(2m-2))$, leads to

$$m \, \text{Var}[X] \trianglelefteq_{\beta(\eta)} \frac{m^2}{m-1} \cdot \left( 1 + \frac{\eta m}{2m-2} \right)^{-1} \text{Var}_m[X], \tag{34}$$

$$\text{where} \quad \beta(\eta) := \eta + \frac{\eta^2 m}{2m-2}.$$

**Proof of Theorem 14.** Let $\eta \in ]0, 1/b[$ and $s_\eta := \eta \cdot \kappa(\eta b)$. We define

$$X_h(Z_i) := \ell_h(Z_i) - \ell_{\hat{h}(Z_{>m})}(Z_i), \quad \text{for } 1 \leq i \leq m,$$

$$\tilde{X}_h(Z_j) := \ell_h(Z_j) - \ell_{\hat{h}(Z_{\leq m})}(Z_j), \quad \text{for } m < j \leq n.$$

Since $\ell$ is bounded from above by $b$, the Bernstein inequality (17) applied to the zero-mean random variables $\mathbb{E}_{Z_i' \sim \mathbf{D}}[X_h(Z_i')] - X_h(Z_i), i \in [n]$, implies that for all $h \in \mathcal{H}$,

$$Y_h^\eta(Z_i) := \mathbb{E}_{Z_i' \sim \mathbf{D}}[X_h(Z_i')] - X_h(Z_i) - s_\eta \cdot \text{Var}[X_h(Z)] \trianglelefteq_\eta 0, \quad \text{for } 1 \leq i \leq m,$$

$$\tilde{Y}_h^\eta(Z_j) := \mathbb{E}_{Z_j' \sim \mathbf{D}}[\tilde{X}_h(Z_j')] - \tilde{X}_h(Z_j) - s_\eta \cdot \text{Var}[\tilde{X}_h(Z)] \trianglelefteq_\eta 0, \quad \text{for } m < j \leq n.$$

Since $Z_1, \ldots, Z_n$ are i.i.d. we can chain the ESIs above using Proposition 10-(b) to get:

$$S := \sum_{i=1}^{m} Y_h^\eta(Z_i) \trianglelefteq_\eta 0, \quad \tilde{S} := \sum_{j=m+1}^{n} \tilde{Y}_h^\eta(Z_j) \trianglelefteq_\eta 0.$$

Chaining $S \trianglelefteq_\eta 0$ [resp. $\tilde{S} \trianglelefteq_\eta 0$] and (34) with $\text{Var}_m \equiv \text{Var}_{\leq m}$ [resp. $\text{Var}_m \equiv \text{Var}_{>m}$] using Proposition 10-(a), yields,

$$W_h^\eta \trianglelefteq_{\frac{\eta\beta(\eta)}{\eta+\beta(\eta)}} 0 \quad \text{and} \quad \tilde{W}_h^\eta \trianglelefteq_{\frac{\eta\beta(\eta)}{\eta+\beta(\eta)}} 0, \quad \text{where} \tag{35}$$

$$W_h^\eta := \sum_{i=1}^{m} \left( \mathbb{E}_{Z_i' \sim \mathbf{D}}[X_h(Z_i')] - X_h(Z_i) \right) - \frac{s_\eta m^2}{m-1} \cdot \left( 1 + \frac{\eta m}{2m-2} \right)^{-1} \text{Var}_{\leq m}[X_h(Z)],$$

$$\tilde{W}_h^\eta := \sum_{j=m+1}^{n} \left( \mathbb{E}_{Z_j' \sim \mathbf{D}}[X_h(Z_j')] - X_h(Z_j) \right) - \frac{s_\eta m^2}{m-1} \cdot \left( 1 + \frac{\eta m}{2m-2} \right)^{-1} \text{Var}_{>m}[X_h(Z)].$$

Let $\lambda(\eta) \coloneqq \eta\beta(\eta)/(\beta(\eta)+\eta)$. Applying PAC-Bayes (Proposition 11) to $W_h^\eta \trianglelefteq_{\lambda(\eta)} 0$ and $\tilde{W}_h^\eta \trianglelefteq_{\lambda(\eta)} 0$ in (35), with priors $P(Z_{>m})$ and $P(Z_{\leq m})$, respectively, and posterior $P_n = P(Z_{\leq n})$ on $\mathcal{H}$, we get:

$$\mathbb{E}_{h \sim P_n}\left[W_h^\eta\right] - \frac{\mathrm{KL}(P_n \| P(Z_{>m}))}{\lambda(\eta)} \trianglelefteq_{\lambda(\eta)} 0, \quad \mathbb{E}_{h \sim P_n}\left[\tilde{W}_h^\eta\right] - \frac{\mathrm{KL}(P_n \| P(Z_{\leq m}))}{\lambda(\eta)} \trianglelefteq_{\lambda(\eta)} 0.$$

We now apply Proposition 10-(a) to chain these two ESIs, which yields

$$\mathbb{E}_{h \sim P_n}\left[W_h^\eta + \tilde{W}_h^\eta\right] \trianglelefteq_{\frac{\lambda(\eta)}{2}} \frac{\mathrm{KL}(P_n \| P(Z_{>m})) + \mathrm{KL}(P_n \| P(Z_{\leq m}))}{\lambda(\eta)}.$$

With the discrete prior $\pi$ on $\mathcal{G}$, we have for any $\hat{\eta} = \hat{\eta}(Z_{\leq n}) \in \mathcal{G} \subset 1/b \cdot [1/\sqrt{n}, 1[$ (see Proposition 12),

$$\mathbb{E}_{h \sim P_n}\left[W_h^\eta + \tilde{W}_h^\eta\right] \trianglelefteq_{\frac{\lambda(\hat{\eta})}{2}} \frac{\mathrm{COMP}_n}{\lambda(\hat{\eta})} - \frac{2\ln\pi(\hat{\eta})}{\lambda(\hat{\eta})}, \ i.e.,$$

$$n \cdot (L(P_n) - L_n(P_n)) \quad \trianglelefteq_{\frac{\lambda(\hat{\eta})}{2}} \ n \cdot \tilde{c}_{\hat{\eta}} \cdot G_n + \frac{\mathrm{COMP}_n + 2\ln\frac{1}{\pi(\hat{\eta})}}{\lambda(\hat{\eta})} \ +$$

$$\left[\sum_{i=1}^m \left(\mathbb{E}_{Z_i' \sim \mathbf{D}}\left[\ell_{\hat{h}_{>m}}(Z_i')\right] - \ell_{\hat{h}_{>m}}(Z_i)\right) + \sum_{j=m+1}^n \left(\mathbb{E}_{Z_j' \sim \mathbf{D}}\left[\ell_{\hat{h}_{\leq m}}(Z_j')\right] - \ell_{\hat{h}_{\leq m}}(Z_j)\right)\right], \quad (36)$$

where $\hat{h}_{>m} \coloneqq \hat{h}(Z_{>m})$ and $\hat{h}_{\leq m} \coloneqq \hat{h}(Z_{\leq m})$. Let $U_n$ denote the quantity between the square brackets in (36). Using the Bernstein inequality in (17) chained with (34), and Proposition 15, we get for any estimator $\hat{\nu}$ on $\mathcal{G}$:

$$U_n \trianglelefteq_{\lambda(\hat{\nu})} \ n \cdot \tilde{c}_{\hat{\nu}} \cdot \left(\mathrm{Var}_{\leq m}[\ell_{\hat{h}(Z_{>m})}(Z)] + \mathrm{Var}_{>m}[\ell_{\hat{h}(Z_{\leq m})}(Z)]\right) + \frac{\ln\frac{1}{\pi(\hat{\nu})}}{\lambda(\hat{\nu})}. \quad (37)$$

By chaining (36) and (37) using Proposition 10-(a), dividing by $n$, we get:

$$L(P_n) \trianglelefteq_{\frac{n\lambda(\hat{\eta})\lambda(\hat{\nu})}{\lambda(\hat{\eta})+2\lambda(\hat{\nu})}} L_n(P_n) + \tilde{c}_{\hat{\eta}} \cdot G_n + \frac{\mathrm{COMP}_n + 2\ln\frac{1}{\pi(\hat{\eta})}}{\lambda(\hat{\eta}) \cdot n} + \tilde{c}_{\hat{\nu}} \cdot G_n' + \frac{\ln\frac{1}{\pi(\hat{\nu})}}{\lambda(\hat{\nu}) \cdot n}. \quad (38)$$

We now apply Proposition 9 to (38) to obtain the following inequality with probability at least $1 - \delta$:

$$L(P_n) \leq L_n(P_n) + \left[\tilde{c}_{\hat{\eta}} \cdot G_n + \frac{\mathrm{COMP}_n + 2\ln\frac{1}{\pi(\hat{\eta}) \cdot \delta}}{\lambda(\hat{\eta}) \cdot n}\right] + \left\{\tilde{c}_{\hat{\nu}} \cdot G_n' + \frac{\ln\frac{1}{\pi(\hat{\nu}) \cdot \delta}}{\lambda(\hat{\nu}) \cdot n}\right\}. \quad (39)$$

Inequality (6) follows after picking $\hat{\nu}$ and $\hat{\eta}$ to be, respectively, estimators which achieve the infimum over the closer of $\mathcal{G}$ of the quantities between braces and square brackets in (39). $\qquad \square$

## C  Proof of Lemma 2

**Proof.** Throughout this proof, we denote $\hat{h}_{>m} \coloneqq \hat{h}(Z_{>m})$ and $\hat{h}_{\leq m} \coloneqq \hat{h}(Z_{\geq m})$. Let $\delta \in ]0,1[$. Since the sample $Z_{\leq m}$ is independent of $Z_{>m}$, we have

$$\frac{2}{n}\sum_{i=1}^m \ell_{\hat{h}_{>m}}(Z_i)^2 = \mathrm{Var}_{\leq m}[\ell_{\hat{h}_{>m}}(Z)] + \left(\frac{1}{m}\sum_{i=1}^m \ell_{\hat{h}_{>m}}(Z_i)\right)^2. \quad (40)$$

On the other hand, from [27, Theorem 10], we have

$$\mathrm{Var}_{\leq m}[\ell_{\hat{h}_{>m}}(Z)] \leq \frac{2(m-1)}{m}\mathrm{Var}[\ell_{\hat{h}_{>m}}(Z)] + \frac{8\ln\frac{1}{\delta}}{n},$$

$$\overset{|\ell|\leq 1}{\leq} \frac{2(m-1)}{m}L(\hat{h}_{>m}) + \frac{8\ln\frac{1}{\delta}}{n}, \quad (41)$$

with probability at least $1 - \delta$. By Hoeffding's inequality, we also have

$$\left(\frac{1}{m}\sum_{i=1}^m \ell_{\hat{h}_{>m}}(Z_i)\right)^2 \leq 2L(\hat{h}_{>m})^2 + \frac{8\ln\frac{1}{\delta}}{n},$$

$$\overset{|\ell|\leq 1}{\leq} 2L(\hat{h}_{>m}) + \frac{8\ln\frac{1}{\delta}}{n}, \quad (42)$$

with probability at least $1 - \delta$. Combining (40), (41), and (42) together using a union bound, yields

$$\frac{2}{n}\sum_{i=1}^{m}\ell_{\hat{h}_{>m}}(Z_i)^2 \leq \frac{4(n-1)}{n}L(\hat{h}_{>m}) + \frac{16\ln\frac{2}{\delta}}{n}, \tag{43}$$

with probability at least $1 - \delta$. Applying the same argument on the second part of the sample $Z_{>m}$, yields

$$\frac{2}{n}\sum_{j=m+1}^{n}\ell_{\hat{h}_{\leq m}}(Z_i)^2 \leq \frac{4(n-1)}{n}L(\hat{h}_{\leq m}) + \frac{16\ln\frac{2}{\delta}}{n}, \tag{44}$$

with probability at least $1 - \delta$. Applying a union bound, and adding together (43) and (44) then dividing by 2, yields,

$$R_n' \leq \frac{2(n-1)}{n}\left(L(\hat{h}_{\leq m}) + L(\hat{h}_{>m})\right) + \frac{16\ln\frac{4}{\delta}}{n},$$

$$\leq 2\left(L(\hat{h}_{\leq m}) + L(\hat{h}_{>m})\right) + \frac{16\ln\frac{4}{\delta}}{n}, \tag{45}$$

with probability at least $1 - \delta$. Diving (45) by $n$ and applying the square-root yields the desired result. $\qquad\square$

# D  Proofs for Section 6

**Proof of Proposition 9.** Let $Z = X - Y$. For fixed $\eta$, Jensen's inequality yields $\mathbb{E}[Z] \leq 0$. For $\eta = \hat{\eta}$ that is either fixed or itself a random variable, applying Markov's inequality to the random variable $e^{-\hat{\eta}Z}$ yields $Z \leq \frac{\ln\frac{1}{\delta}}{\hat{\eta}}$, with probability at least $1 - \delta$, for any $\delta \in ]0, 1[$. $\qquad\square$

**Proof of Proposition 10.** **[Part (a)]** Fix $(\gamma_i)_{i\in[n]} \in ]0, +\infty[^n$, and let $\nu_j := \left(\sum_{i=1}^{j}\frac{1}{\gamma_i}\right)^{-1}$, for $j \in [n]$. We proceed by induction to show that $\forall j \in [n]$, $\sum_{i=1}^{j}Z_i \trianglelefteq_{\nu_j} 0$. The result holds trivially for $j = 1$, since $\nu_1 = \gamma_1$. Suppose that

$$\sum_{i=1}^{j}Z_i \trianglelefteq_{\nu_j} 0, \tag{46}$$

for some $1 \leq j < n$. We now show that (46) holds for $j + 1$; we have,

$$\mathbb{E}\left[e^{\frac{\nu_j\gamma_{j+1}}{\nu_j+\gamma_{j+1}}\left(\sum_{i=1}^{j}Z_i + Z_{j+1}\right)}\right] = \mathbb{E}\left[e^{\frac{\nu_j\gamma_{j+1}}{\nu_j+\gamma_{j+1}}\sum_{i=1}^{j}Z_i + \frac{\nu_j\gamma_{j+1}}{\nu_j+\gamma_{j+1}}Z_{j+1}}\right],$$

$$\overset{\text{Jensen}}{\leq} \frac{\gamma_{j+1}}{\nu_j+\gamma_{j+1}}\mathbb{E}\left[e^{\nu_j\sum_{i=1}^{j}Z_i}\right] + \frac{\nu_j}{\nu_j+\gamma_{j+1}}\mathbb{E}\left[e^{\gamma_{j+1}Z_{j+1}}\right],$$

$$\overset{\text{using (46)}}{\leq} 1.$$

Thus the result holds for $j + 1$, since $\nu_{j+1} = \frac{\nu_j\gamma_{j+1}}{\nu_j+\gamma_{j+1}}$. This establishes (14).

**[Part (b)]** This is a special case of [22, Lemma 6], who treat the general case with non-i.i.d. distributions. $\qquad\square$

**Proof of Proposition 11.** Let $\rho(h) = (dP_n/dP_0)(h)$ be the density of $h \in \mathcal{H}$ relative to the prior measure $P_0$. We then have $\mathrm{KL}(P_n\|P_0) = \mathbb{E}_{h\sim P_n}[\ln\rho(h)]$. We can now write:

$$\mathbb{E}\left[e^{\eta\mathbb{E}_{h\sim P_n}[Y_h]-\mathrm{KL}(P_n\|P_0)}\right] = \mathbb{E}\left[e^{\eta\mathbb{E}_{h\sim P_n}[Y_h-\ln\rho(h)]}\right],$$

$$\leq \mathbb{E}\left[\mathbb{E}_{h\sim P_n}\left[e^{\eta(Y_h-\ln\rho(h))}\right]\right], \quad \text{(Jensen's Inequality)}$$

$$= \mathbb{E}\left[\mathbb{E}_{h\sim P_n}\left[\frac{dP_0}{dP_n}\cdot e^{\eta Y_h}\right]\right],$$

$$= \mathbb{E}\left[\mathbb{E}_{h\sim P_0}\left[e^{\eta Y_h}\right]\right],$$

$$= \mathbb{E}_{h\sim P_0}\left[\mathbb{E}\left[e^{\eta Y_h}\right]\right], \quad \text{(Tonelli's Theorem)}$$

$$= 1,$$

where the final equality follows from our assumption that $Y_h \trianglelefteq_\eta 0$, for all $h \in \mathcal{H}$. $\qquad\square$

**Proof of Proposition 12.** Since $Y_\eta \unlhd_\eta 0$, for $\eta \in \mathcal{G}$, we have in particular:

$$1 \geq \mathbb{E}\left[\sum_{\eta \in \mathcal{G}} \pi(\eta)e^{\eta Y_\eta}\right] \geq \mathbb{E}\left[\pi(\hat{\eta})e^{\hat{\eta}Y_{\hat{\eta}}}\right], \tag{47}$$

where the right-most inequality follows from the fact that the expectation of a countable sum of positive random variable is greater than the expectation of a single element in the sum. Rearranging (47) gives (15). □

## E Proof of Theorem 7

In what follows, for $h \in \mathcal{H}$, we denote $X_h(Z) \coloneqq \ell_h(Z) - \ell_{h_*}(Z)$ the excess loss random variable, where $h_*$ is the risk minimizer within $\mathcal{H}$. Let

$$\rho(\eta) \coloneqq \frac{1}{\eta} \ln \mathbb{E}_{Z \sim \mathbf{D}}\left[e^{-\eta X_h(Z)}\right]$$

be its *normalized cumulant generating function*. We need the following useful lemmas:

**Lemma 15. [22]** *Let $h \in \mathcal{H}$ and $X_h$ be as above. Then, for all $\eta \geq 0$,*

$$\alpha_\eta \cdot X_h(Z)^2 - X_h(Z) \unlhd_\eta \rho(2\eta) + \alpha_\eta \cdot \rho(2\eta)^2, \quad \text{where } \alpha_\eta \coloneqq \frac{\eta}{1 + \sqrt{1 + 4\eta^2}}.$$

**Lemma 16. [22]** *Let $b > 0$, and suppose that $X_h \in [-b, b]$ almost surely, for all $h \in \mathcal{H}$. If the $(\beta, B)$-Bernstein condition holds with $\beta \in [0, 1]$ and $B > 0$, then*

$$\rho(\eta) \leq (B\eta)^{\frac{1}{1-\beta}}, \quad \text{for all } \eta \in ]0, 1/b].$$

**Lemma 17. [12]** *Let $b > 0$, and suppose that $X_h \in [-b, b]$ almost surely, for all $h \in \mathcal{H}$. Then*

$$\rho(\eta) \leq \frac{\eta b^2}{2}, \quad \text{for all } \eta \in \mathbb{R}.$$

**Proof of Theorem 7.** First we apply the following inequality

$$(a - d)^2 \leq 2(a - c)^2 + 2(d - c)^2 \tag{48}$$

which holds for all $a, c, d \in \mathbb{R}$ to upper bound $V_n$. Let's focus on the first term in the expression of $V_n$, which we denote $V_n^{\text{left}}$: that is,

$$V_n^{\text{left}} \coloneqq \mathbb{E}_{h \sim P_n}\left[\frac{1}{n}\sum_{i=1}^{m}\left(\ell_h(Z_i) - \mathbb{E}_{h' \sim Q(Z_{>i})}[\ell_{h'}(Z_i)]\right)^2\right].$$

Letting $X_h(Z) \coloneqq \ell_h(Z) - \ell_{h_*}(Z)$ and applying (48) with $a = \ell_h(Z_i)$, $c = \ell_{h_*}(Z_i)$, and $d = \mathbb{E}_{h' \sim Q(Z_{>i})}[\ell_{h'}(Z_i)] \stackrel{*}{=} \mathbb{E}_{h' \sim Q(Z_{>m})}[\ell_{h'}(Z_i)]$ (where $\stackrel{*}{=}$ is due to our assumption on $Q$), we get:

$$V_n^{\text{left}} \leq \mathbb{E}_{h \sim P_n}\left[\frac{2}{n}\sum_{i=1}^{m}X_h(Z_i)^2\right] + \frac{2}{n}\sum_{i=1}^{m}\left(\mathbb{E}_{h' \sim Q(Z_{>m})}[\ell_{h'}(Z_i)] - \ell_{h_*}(Z_i)\right)^2,$$

$$\leq \mathbb{E}_{h \sim P_n}\left[\frac{2}{n}\sum_{i=1}^{m}X_h(Z_i)^2\right] + \mathbb{E}_{h \sim Q(Z_{>m})}\left[\frac{2}{n}\sum_{i=1}^{m}X_h(Z_i)^2\right]. \quad \text{(by Jensen's Inequality) (49)}$$

Let $i \in [m]$, $h \in \mathcal{H}$, and $\eta \in ]0, 1/b[$. Under the $(\beta, B)$-Bernstein condition, Lemmas 15-17 imply,

$$\alpha_\eta \cdot X_h(Z_i)^2 \unlhd_\eta X_h(Z_i) + \left(1 + \frac{b}{2}\right)(2B\eta)^{\frac{1}{1-\beta}}, \tag{50}$$

where $\alpha_\eta \coloneqq \eta/(1 + \sqrt{1 + 4\eta^2})$. Now, due to the Bernstein inequality (17), we have

$$X_h(Z_i) \unlhd_\eta \mathbb{E}_{Z'_i \sim \mathbf{D}}[X_h(Z'_i)] + s_\eta \cdot \mathbb{E}_{Z'_i \sim \mathbf{D}}[X_h(Z'_i)^2], \quad \text{where } s_\eta \coloneqq \eta \cdot \kappa(\eta b),$$

$$\unlhd \mathbb{E}_{Z'_i \sim \mathbf{D}}[X_h(Z'_i)] + s_\eta \cdot \mathbb{E}_{Z'_i \sim \mathbf{D}}[X_h(Z'_i)]^\beta, \quad \text{(by the Bernstein condition)}$$

$$\unlhd_\eta 2\mathbb{E}_{Z'_i \sim \mathbf{D}}[X_h(Z'_i)] + a_\beta^{\frac{\beta}{1-\beta}} \cdot (s_\eta)^{\frac{1}{1-\beta}}, \quad \text{where } a_\beta \coloneqq (1 - \beta)^{1-\beta}\beta^\beta. \tag{51}$$

The last inequality follows by the fact that $z^\beta = a_\beta \cdot \inf_{\nu>0}\{z/\nu + \nu^{\frac{\beta}{1-\beta}}\}$, for $z \geq 0$ (in our case, we set $\nu = a_\beta \cdot s_\eta$ to get to (51)). By chaining (50) with (51) using Proposition 10-(a), we get:

$$\alpha_\eta \cdot X_h(Z_i)^2 \trianglelefteq_{\frac{\eta}{2}} 2\mathbb{E}_{Z_i'\sim\mathbf{D}}\left[X_h(Z_i')\right] + a_\beta^{\frac{\beta}{1-\beta}} \cdot (s_\eta)^{\frac{1}{1-\beta}} + \left(1 + \frac{b}{2}\right)(2B\eta)^{\frac{1}{1-\beta}}.$$

$$\trianglelefteq_{\frac{\eta}{2}} 2\mathbb{E}_{Z_i'\sim\mathbf{D}}\left[X_h(Z_i')\right] + \mathcal{P}\cdot\eta^{\frac{1}{1-\beta}}, \text{ with } \mathcal{P} \coloneqq a_\beta^{\frac{\beta}{1-\beta}} + \left(1 + \frac{b}{2}\right)(2B)^{\frac{1}{1-\beta}}, \quad (52)$$

where in the last inequality we used $\kappa(1) \leq 1$. Since (52) holds for all $h \in \mathcal{H}$, it still holds in expectation over $\mathcal{H}$ with respect to the distribution $Q(Z_{>m})$ (recall that $i \leq m$);

$$\alpha_\eta \cdot \mathbb{E}_{h\sim Q(Z_{>m})}\left[X_h(Z_i)^2\right] \trianglelefteq_{\frac{\eta}{2}} 2\mathbb{E}_{h\sim Q(Z_{>m})}\left[\mathbb{E}_{Z_i'\sim\mathbf{D}}\left[X_h(Z_i')\right]\right] + \mathcal{P}\cdot\eta^{\frac{1}{1-\beta}}. \quad (53)$$

Since the samples $Z_{\leq n}$ are i.i.d, we have $\mathbb{E}_{Z_i\sim\mathbf{D}}\left[\ell_h(Z_i)\right] = \mathbb{E}_{Z_j\sim\mathbf{D}}\left[\ell_h(Z_j)\right]$, for all $i,j \in [m]$. Thus, after summing (52) and (53), for $i = 1, \ldots, m$, using Proposition 10-(b) and dividing by $n$, we get

$$\frac{\alpha_\eta}{n}\sum_{i=1}^m X_h(Z_i)^2 \trianglelefteq_{\frac{n\cdot\eta}{2}} \mathbb{E}_{Z\sim\mathbf{D}}\left[X_h(Z)\right] + \frac{\mathcal{P}}{2}\cdot\eta^{\frac{1}{1-\beta}}, \quad (54)$$

$$\mathbb{E}_{h\sim Q(Z_{>m})}\left[\frac{\alpha_\eta}{n}\sum_{i=1}^m X_h(Z_i)^2\right] \trianglelefteq_{\frac{n\cdot\eta}{2}} \mathbb{E}_{h\sim Q(Z_{>m})}\left[\mathbb{E}_{Z\sim\mathbf{D}}\left[X_h(Z)\right]\right] + \frac{\mathcal{P}}{2}\cdot\eta^{\frac{1}{1-\beta}}. \quad (m = n/2)(55)$$

Now we apply PAC-Bayes (Proposition 11) to (54), with prior $P(Z_{>m})$ and posterior $P_n$, and obtain:

$$\mathbb{E}_{h\sim P_n}\left[\frac{\alpha_\eta}{n}\sum_{i=1}^m X_h(Z_i)^2\right] \trianglelefteq_{\frac{n\cdot\eta}{2}} \mathbb{E}_{h\sim P_n}\left[\mathbb{E}_{Z\sim\mathbf{D}}\left[X_h(Z)\right]\right] + \frac{\mathcal{P}}{2}\cdot\eta^{\frac{1}{1-\beta}} + \frac{2\mathrm{KL}(P_n\|P(Z_{>m}))}{\eta\cdot n}. (56)$$

Note that the upper-bound on $V_n^{\text{left}}$ in (49) is the sum of the left-hand sides of (55) and (56) divided by $\alpha_\eta/2$. From now on, we restrict $\eta$ to the range $]0, 1/(2b)[$ and define

$$\mathcal{A}_\eta \coloneqq \frac{2c_\eta}{\alpha_\eta} \leq 2\vartheta\left(\frac{1}{2}\right)\cdot\left(1 + \sqrt{1 + \frac{1}{b^2}}\right) =: \mathcal{A}, \quad \eta \in \left]0, \frac{1}{2b}\right[.$$

Chaining (55) and (56) using Proposition 10-(a) and multiplying throughout by $\mathcal{A}_\eta$, yields

$$c_\eta \cdot V_n^{\text{left}} \trianglelefteq_{\frac{n\eta}{4\mathcal{A}_\eta}} \mathcal{A}\cdot\left(\bar{L}(P_n) + \bar{L}(Q(Z_{>m}))\right) + \mathcal{P}\mathcal{A}\eta^{\frac{1}{1-\beta}} + \frac{2\mathcal{A}\cdot\mathrm{KL}(P_n\|P(Z_{>m}))}{\eta\cdot n}. \quad (57)$$

By a symmetric argument, a version of (57), with $Q(Z_{>m})$ [resp. $P(Z_{>m})$] replaced by $Q(Z_{\leq m})$ [resp. $P(Z_{\leq m})$], holds for $V_n^{\text{right}} \coloneqq V_n - V_n^{\text{left}}$. Using Proposition 10-(a) again, to chain the ESI inequalities of $c_\eta \cdot V_n^{\text{left}}$ and $c_\eta \cdot V_n^{\text{right}}$, we obtain:

$$c_\eta \cdot V_n \trianglelefteq_{\frac{n\eta}{8\mathcal{A}_\eta}} \mathcal{A}\cdot\left(2\bar{L}(P_n) + \bar{L}(Q_{\leq m}) + \bar{L}(Q_{>m})\right) + 2\mathcal{P}\mathcal{A}\eta^{\frac{1}{1-\beta}} + \frac{2\mathcal{A}\cdot\mathrm{COMP}_n}{\eta\cdot n}, \quad (58)$$

where $Q_{>m} \coloneqq Q(Z_{>m})$ and $Q_{\leq m} \coloneqq Q(Z_{\leq m})$. Let $\delta \in ]0,1[$, and $\pi$ and $\mathcal{G}$ be as in (8). Applying Proposition 12 to (58) to obtain the corresponding ESI inequality with a random estimator $\hat{\eta} = \hat{\eta}(Z_{\leq n})$ with support on $\mathcal{G}$, and then applying Proposition 9, we get, with probability at least $1 - \delta$,

$$c_{\hat{\eta}} \cdot V_n \leq \mathcal{A}\cdot\left(2\bar{L}(P_n) + \bar{L}(Q_{\leq m}) + \bar{L}(Q_{>m})\right) + 2\mathcal{P}\mathcal{A}\hat{\eta}^{\frac{1}{1-\beta}} + \frac{2\mathcal{A}\cdot\mathrm{COMP}_n + 8\mathcal{A}\ln\frac{|\mathcal{G}|}{\delta}}{\hat{\eta}\cdot n}. \quad (59)$$

Now adding $(\mathrm{COMP}_n + \varepsilon_{\delta,n})/(\hat{\eta}\cdot n)$ on both sides of (59) and choosing the estimator $\hat{\eta}$ optimally in the closure of $\mathcal{G}$ yields the desired result. $\qquad\square$

## F  Proof of Lemma 13

**Proof.** Part (a) of the lemma was shown in the main body of the paper[2]. Thus, we only prove part (b); we will show a slight extension, namely that for all $0 < u < 1$, for all $\beta > 0, u > 0$,

$$\sup_{\rho\leq u} \sup_{P:\mathbb{E}_P[X]=\rho, P(X\leq u)=1} \mathbb{E}_{X\sim P}\left[e^{\beta E[X]-X-cX^2}\right] > 1 \text{ if } 0 < c < \vartheta(u) \text{ or } \beta \neq 1.$$

The statement of the lemma (16) follows as the special case for $\beta = 1$, by replacing $X$ by $\eta X$ and setting $u$ to $u := \eta b < 1$.

We prove this by considering the set of distributions satisfying the constraint $\mathbb{E}[X] = \rho$ that are supported on at most two points,

$$\mathcal{P}_{\underline{x}, \rho, \bar{x}, u} = \{P : P\{\underline{x}\} + P\{\bar{x}\} = 1; \mathbb{E}_P[X] = \rho, \underline{x} \le \bar{x} \le u\},$$

and showing that

$$\sup_{\rho \le u} \sup_{P \in \mathcal{P}_{\underline{x}, \rho, \bar{x}, u}} g_{c,\beta}(P), \text{ with } g_{c,\beta}(P) := \mathbb{E}_{X \sim P}\left[e^{\beta \rho - X - cX^2}\right]$$

is larger than 1. We first show that , for any $\beta \neq 1$, we can choose such a $P$ such that $\sup_{P \in \mathcal{P}_{\underline{x}, \rho, \bar{x}, u}} g_{c,\beta}(P) > 1$. To see this, write $g_{c,\beta}(P)$ as

$$p \cdot e^{-\underline{x} + \beta \rho - c\underline{x}^2} + (1-p)e^{-\bar{x} + \beta \rho - c\bar{x}^2}$$

with $\rho = \mathbb{E}_P[X]$. We need to maximize this over $\rho = p\underline{x} + (1-p)\bar{x}$, so that in the end, we want to maximize over $0 \le p \le 1, \underline{u} \le \underline{x} \le \bar{x} \le u$, the expression

$$p \cdot e^{-\underline{x} + \beta(p\underline{x} + (1-p)\bar{x}) - c\underline{x}^2} + (1-p)e^{-\bar{x} + \beta(p\underline{x} + (1-p)\bar{x}) - c\bar{x}^2}$$

Now we write $\underline{x} = \bar{x} - a$ for some $a \ge 0$. The expression becomes

$$p \cdot e^{-\beta pa + (\beta - 1)\bar{x} + a - c(\bar{x} - a)^2} + (1-p) \cdot e^{-\beta pa + (\beta - 1)\bar{x} - c\bar{x}^2}$$

which is equal to

$$f(p, a, \bar{x}) := e^{-c\bar{x}^2 - \beta pa + (\beta - 1)\bar{x}}\left(pe^{a + 2ca\bar{x} - ca^2} + 1 - p\right) \underset{\text{if } \beta = 1}{=} e^{c\bar{x}^2 - pa}\left(pe^{a + 2ca\bar{x} - ca^2} + 1 - p\right),$$

where the dependency of $f$ on $c$ and $\beta$ is suppressed in the notation. At $p = 1$ and $p = 0$, this simplifies to (using also $\underline{x}$ again)

$$f(1, a, \bar{x}) = e^{-c\bar{x}^2 - \beta a + (\beta - 1)\bar{x}}\left(e^{a + 2ca\bar{x} - ca^2}\right) = e^{-c\underline{x}^2 + (\beta - 1)\underline{x}} \underset{\text{if } \beta = 1}{=} e^{-c\underline{x}^2}$$

$$f(0, a, \bar{x}) = e^{-c\bar{x}^2 + (\beta - 1)\bar{x}} \underset{\text{if } \beta = 1}{=} e^{-c\bar{x}^2}.$$

If $\beta < 1$, we can choose $\underline{x} = \bar{x} - a$ negative yet very close to 0 making $f(1, a, \bar{x}) > 1$; if $\beta > 1$, we can choose $\bar{x}$ positive yet very close to 0 making $f(0, a, \bar{x}) > 1$. Thus, $\sup g_{c,\beta}(P)$ can be made larger than 1 by $P$ satisfying the constraint if $\beta \neq 1$. This shows (F) for the case $\beta \neq 1$. Hence, from now on we restrict to the case $\beta = 1$; we will further restrict to $\underline{x}$ and $\bar{x}$ such that $\underline{x} \le 0 \le \bar{x}$ so $\bar{x} \le a$. We will determine the maximum over (F) for $a \ge \bar{x}$ and $0 \le p \le 1$, for each given $0 \le \bar{x} \le u$. The partial derivatives to $p$ and $a$ are:

$$\frac{\partial}{\partial p}f(p, a, \bar{x}) = e^{-c\bar{x}^2 - pa}\left(\left(e^{a + 2ca\bar{x} - ca^2} - 1\right) - a \cdot \left(pe^{a + 2ca\bar{x} - ca^2} + (1 - p)\right)\right)$$

$$= e^{-c\bar{x}^2 - pa}\left(e^{a + 2ca\bar{x} - ca^2}(1 - ap) - 1 - a + ap\right)$$

$$\frac{\partial}{\partial a}f(p, a, \bar{x}) = -p \cdot e^{-c\bar{x}^2 - pa}\left(pe^{a + 2ca\bar{x} - ca^2} + (1 - p)\right) +$$

$$+ e^{-c\bar{x}^2 - pa} \cdot p \cdot e^{a + 2ca\bar{x} - ca^2} \cdot (1 + 2c\bar{x} - 2ca)$$

$$= p(1 - p) \cdot e^{-c\bar{x}^2 - pa} \cdot \left(-1 + e^{a + 2ca\bar{x} - ca^2}(1 + 2c\frac{\bar{x} - a}{1 - p})\right).$$

At $a = \bar{x}$ (i.e. $\underline{x} = 0$), $f(p, a, \bar{x})$ simplifies to

$$f(p, \bar{x}, \bar{x}) = e^{-c\bar{x}^2 - p\bar{x}} \cdot (pe^{\bar{x} + c\bar{x}^2} + (1 - p)) \text{ so } f(1, \bar{x}, \bar{x}) = 1$$

and the partial derivative to $p$ at $(p, a, \bar{x}) = (1, \bar{x}, \bar{x})$ becomes

$$e^{-c\bar{x}^2 - \bar{x}}\left(\left(e^{\bar{x} + c(\bar{x})^2} - 1\right) - \bar{x}e^{\bar{x} + c(\bar{x})^2}\right) = 1 - e^{-c\bar{x}^2 - \bar{x}} - \bar{x}.$$

If (F) is negative, we can take $a = \bar{x}$ and $p$ slightly smaller than 1 to get $f(p, a, \bar{x}) > 1$. This happens if and only if $c$ is smaller than

$$\frac{-\ln(1 - \bar{x}) - \bar{x}}{\bar{x}^2} = \vartheta(\bar{x}).$$

Thus, by taking $\underline{x} = 0$ and $\bar{x} = a = u$, and $p$ slightly smaller than 1 again, we get $f(p, a, \bar{x}) > 1$ if $c < \vartheta(u)$; this shows (F) for the case $\beta = 1$; the result is proved. $\qquad\square$

# G   Comparison Between "Bernstein" Inequalities

**Discussion and Proof of Our Version of Bernstein's Inequality (17).**   Standard versions of Bernstein's inequality (see [12], and [15, Lemma 5.6]) can also be brought in ESI notation. In particular, compared with our version they express the inequality in terms of the random variable $Y = -X$, which is then upper bounded by $b$; more importantly, they have the second moment rather than the variance on the right-hand side, resulting in a slightly worse multiplicative factor $\kappa(2\eta b)$ instead of our $\kappa(\eta b)$; the proof is a standard one (see [12, Lemma A.4]) with trivial modifications: let $U := \eta X$ and $\bar{u} := \eta b$. Since $\kappa(u)$ is nondecreasing in $u$ and $U \le \bar{u}$, we have

$$\frac{e^U - U - 1}{U^2} \le \frac{e^{\bar{u}} - \bar{u} - 1}{\bar{u}^2},$$

and hence $e^U - U - 1 \le \kappa(\bar{u})U^2$. Taking expectation on both sides and using that $\ln \mathbb{E}[e^U] \le \mathbb{E}[U] - 1$, we get $\ln \mathbb{E}[e^U] - \mathbb{E}[U] \le \kappa(\bar{u})\mathbb{E}[U^2]$. The result follows by exponentiating, rearranging, and using the ESI definition.

**Comparison Between Un-expected and Empirical Bernstein Inequalities.**   The proof of the following proposition demonstrates how the un-expected Bernstein inequality in Lemma 13 together with the standard Bernstein inequality (17) imply a version of the empirical Bernstein inequality in [27, Theorem 4] with slightly worse factors. However, the latter inequality cannot be used to derive our main result — we do really require our new inequality to show Theorem 3, since we need to "chain" it to work with samples of length $n$ rather than 1 in a different way. In the next proposition, we will use the following grid $\mathcal{G}$ and distribution $\pi$,

$$\mathcal{G} := \left\{ \frac{1}{\nu}, \ldots, \frac{1}{\nu^K} : K := \left\lceil \log_\nu \left( \sqrt{\frac{n}{2 \ln \frac{2}{\delta}}} \right) \right\rceil \right\}, \quad \text{and} \quad \pi = \text{uniform distribution over } \mathcal{G}. \tag{60}$$

for $\nu > 0$. To simplify the presentation, we will use $\nu = 2$ in the next proposition, albeit this may not be the optimal choice.

**Proposition 18.** *Let $\mathcal{G}$ be as in (60) with $\rho = 2$, and $Z, Z_1, \ldots, Z_n$ be i.i.d random variables taking values in $[0, 1]$. Then, for all $\delta \in \,]0, 1[$, with probability at least $1 - \delta$,*

$$\mathbb{E}[Z] - \frac{1}{n}\sum_{i=1}^{n} Z_i \le \left( 3\sqrt{\frac{\mathrm{Var}_n[Z] \cdot \ln \frac{2|\mathcal{G}|}{\delta}}{2n}} + \frac{11 \ln \frac{2|\mathcal{G}|}{\delta}}{10n} \right) \vee \frac{11 \ln \frac{2|\mathcal{G}|}{\delta}}{4n} + \frac{c_{1/2} \cdot \ln \frac{2}{\delta}}{2n},$$

*where $\mathrm{Var}_n[Z] := \frac{1}{n}\sum_{i=1}^{n} \left( Z_i - \frac{1}{n}\sum_{j=1}^{n} Z_j \right)^2$ is the empirical variance, $c_{1/2} := \vartheta(1/2)/2$, and $\vartheta$ as in Lemma 13.*

**Proof.**   Let $\delta \in \,]0, 1[$. Applying Lemma 13 to $X_i = Z_i - \mathbb{E}[Z]$, for $i \in [n]$, we get, for all $0 < \eta < 1/2$,

$$\mathbb{E}[Z] - Z_i \trianglelefteq_\eta c_\eta \cdot (Z_i - \mathbb{E}[Z])^2, \quad \text{where } c_\eta := \eta \cdot \vartheta(\eta). \tag{61}$$

Applying Proposition 10-(b) to chain (61) for $i = 1, \ldots, n$, then dividing by $n$ yields

$$\mathbb{E}[Z] - \frac{1}{n}\sum_{i=1}^{n} Z_i \trianglelefteq_{n\eta} \frac{c_\eta}{n}\sum_{i=1}^{n} (Z_i - \mathbb{E}[Z])^2,$$

$$= c_\eta \cdot \mathrm{Var}_n[Z] + c_\eta \cdot \left( \mathbb{E}[Z] - \frac{1}{n}\sum_{i=1}^{n} Z_i \right)^2, \tag{62}$$

where the equality follows from the standard bias-variance decomposition. Let $\mathcal{G}$ and $\pi$ be as in (60), and let $\hat{\eta} = \hat{\eta}(Z_{\le n})$ be any random estimator with support on $\mathcal{G}$. By Proposition 12, a version of (62) with $\eta$ is replaced by $\hat{\eta}$ and $\ln(|\mathcal{G}|)/(n\hat{\eta})$ added to its RHS also holds. By applying Proposition 9 to this new inequality, we get, with probability at least $1 - \delta$,

$$\mathbb{E}[Z] - \frac{1}{n}\sum_{i=1}^{n} Z_i \le c_{\hat{\eta}} \cdot \mathrm{Var}_n[Z] + \frac{\ln \frac{|\mathcal{G}|}{\delta}}{n \cdot \hat{\eta}} + c_{\hat{\eta}} \cdot \left( \mathbb{E}[Z] - \frac{1}{n}\sum_{i=1}^{n} Z_i \right)^2. \tag{63}$$

Now using Hoeffding's inequality [27, Theorem 3], we also have

$$\left(\mathbb{E}[Z] - \frac{1}{n}\sum_{i=1}^{n} Z_i\right)^2 \le \frac{\ln\frac{1}{\delta}}{2n}, \tag{64}$$

with probability at least $1 - \delta$. Thus, by combining (63) and (64) via the union bound, we get that, with probability at least $1 - \delta$,

$$\mathbb{E}[Z] - \frac{1}{n}\sum_{i=1}^{n} Z_i \le \left(c_{\hat{\eta}} \cdot \mathrm{Var}_n[Z] + \frac{\ln\frac{2|\mathcal{G}|}{\delta}}{n \cdot \hat{\eta}}\right) + \frac{c_{\hat{\eta}} \cdot \ln\frac{2}{\delta}}{2n}. \tag{65}$$

We now use the fact that for all $\eta \in ]0, 1/2[$,

$$c_\eta = \eta \cdot \vartheta(\eta) \le \frac{\eta}{2} + \frac{11\eta^2}{20}. \tag{66}$$

Let $\hat{\eta}_* \in [0, +\infty]$ be the un-constrained estimator defined by

$$\hat{\eta}_* := \sqrt{\frac{2\ln\frac{2|\mathcal{G}|}{\delta}}{\mathrm{Var}_n[Z] \cdot n}}.$$

Note that by our choice of $\mathcal{G}$ in (60), we always have $\hat{\eta}_* \ge \min \mathcal{G}$. Let $\hat{\eta} \in ([\hat{\eta}_*/2, \hat{\eta}_*] \cap \mathcal{G}) \ne \varnothing$, if $\hat{\eta}_* \le 1$, and $\hat{\eta} = 1/2$, otherwise. In the first case (*i.e.* when $\hat{\eta}_* \le 1$), substituting $\eta$ for $\hat{\eta} \in ([\hat{\eta}_*/2, \hat{\eta}_*] \cap \mathcal{G}$ in the expression between brackets in (65), and using the fact that $\hat{\eta}_*/2 \le \hat{\eta} \le \hat{\eta}_*$ and (66), gives

$$c_{\hat{\eta}} \cdot \mathrm{Var}_n[Z] + \frac{\ln\frac{2|\mathcal{G}|}{\delta}}{\hat{\eta} \cdot n} \le (1 + 2)\sqrt{\frac{\mathrm{Var}_n[Z] \cdot \ln\frac{2|\mathcal{G}|}{\delta}}{2n}} + \frac{11 \cdot \ln\frac{2|\mathcal{G}|}{\delta}}{10n}. \tag{67}$$

Now for the case where $\hat{\eta}_* \ge 1$, we substitute $\eta$ for $\hat{\eta} = 1/2$ in the expression between brackets in (65), and use (66) and the fact that $1 \le \hat{\eta}_* = \sqrt{2\ln(2|\mathcal{G}|/\delta)/(\mathrm{Var}_n[Z] \cdot n)}$, we get:

$$
\begin{aligned}
c_{\hat{\eta}} \cdot \mathrm{Var}_n[Z] + \frac{\ln\frac{2|\mathcal{G}|}{\delta}}{\hat{\eta} \cdot n} &\le \left(\frac{\hat{\eta}}{2} + \frac{11\hat{\eta}^2}{20}\right) \cdot \mathrm{Var}_n[Z] + \frac{2 \cdot \ln\frac{2|\mathcal{G}|}{\delta}}{n}, \\
&\le \left(\frac{\hat{\eta}}{2} + \frac{11\hat{\eta}^2}{20}\right) \cdot \frac{2\ln\frac{2|\mathcal{G}|}{\delta}}{n} + \frac{2 \cdot \ln\frac{2|\mathcal{G}|}{\delta}}{n}, \quad \text{(due to } \hat{\eta}_* \ge 1) \\
&= \frac{11\ln\frac{2|\mathcal{G}|}{\delta}}{4n}, \quad (\hat{\eta} = 1/2)
\end{aligned}
\tag{68}
$$

Combining (65), with (67) and (68) yields the desired results. □

## H   Additional Experiments

### H.1   Informed Priors

In this section, we run the same experiments as in Section 4 of the main body, except for the following changes

- For Maurer's bound, we use the version in our Lemma 4 with informed priors.
- For the TS and Catoni bounds, we build a prior from the first half of the data (*i.e.* we replace $P_0$ by $P(Z_{\le m})$, where $m = n/2$) and use it to evaluate the bounds on the second half of the data. In this case, the "posterior" distribution is $P(Z_{>m})$, and thus the term $\mathrm{KL}(P_n\|P_0)$ is replaced by $\mathrm{KL}(P(Z_{>m})\|P(Z_{\le m}))$.

Recall that $P(Z_{>m}) \equiv \mathcal{N}(\hat{h}(Z_{>m}), \sigma^2 I_d)$, $P(Z_{\le m}) \equiv \mathcal{N}(\hat{h}(Z_{\le m}), \sigma^2 I_d)$, and $P(Z_{\le n}) \equiv \mathcal{N}(\hat{h}(Z_{\le n}), \sigma^2 I_d)$, where the variance $\sigma^2$ is learned from a geometric grid (see Section 4); our own bound is not affected by any of these changes. The results for the synthetic and UCI datasets are reported in Figure 2 and Table 2, respectively.

Figure 2: Results for the synthetic data with informed priors.

| Dataset | n | d | Test error of $\hat{h}$ | Our | Maurer | TS | Catoni |
|---------|------|-----|-------------|-------|--------|-------|--------|
| Haberman | 244 | 3 | 0.272 | 0.52 | 0.459 | 0.501 | 0.55 |
| Breast-C. | 560 | 9 | 0.068 | 0.185 | 0.164 | 0.215 | 0.219 |
| Tic-Tac-Toe | 766 | 27 | 0.046 | 0.19 | 0.152 | 0.202 | 0.199 |
| Bank-note | 1098 | 4 | 0.058 | 0.125 | 0.117 | 0.136 | 0.143 |
| kr-vs-kp | 2556 | 73 | 0.044 | 0.107 | 0.102 | 0.123 | 0.127 |
| Spam-base | 3680 | 57 | 0.173 | 0.293 | 0.284 | 0.317 | 0.323 |
| Mushroom | 6500 | 116 | 0.002 | 0.018 | 0.016 | 0.023 | 0.024 |
| Adult | 24130 | 108 | 0.168 | 0.195 | 0.198 | 0.2 | 0.203 |

Table 2: Results for the UCI datasets.

Though our bound still performs better than Catoni's and TS, Maurer's bound in Lemma 4 tends to be slightly tighter than ours, especially when the sample size is small. We note, however, that the advantage of our bound has not been fully leveraged here; our bound in its full generality in Theorem 3 allows one to use "online posteriors" $(Q(Z_{>i}))$ and $(Q(Z_{<j}))$ in the $V_n$ term which converge to the one based on the full sample, *i.e.* $Q(Z_{\leq n})$. We expect this to substantially improve our bound. However, we did not experiment with this due to computational reasons.

## H.2 Maurer's Bound: Informed Versus Uninformed Priors

In this section, we compare the performance of Maurer's bound with and without informed priors (*i.e.* (2) and (9), respectively) on synthetic data in the same setting as Section 4. From Figure 3, we see that using informed priors as in Lemma 4 substantially improves Maurer's bound.

## H.3 Varying the Bayes Error and Bayes Act

In this subsection, we run the same synthetic experiment as in Subsection (H.1) (*i.e.* using informed priors for all bounds), except for the following changes:

- We vary the Bayes error by varying the level of noise: we flip the labels with probability either 0.05, 0.1, or 0.2 (note that in Section 4 we flipped labels with probability 0.1).

Figure 3: Results for the synthetic data: (Blue curve) Uninformed Maurer's bound (2); (Red curve) Informed Maurer's bound (9).

Figure 4: Results for the synthetic data with informed priors, randomly generated Bayes act, and Bayes error set to 0.05.

Figure 5: Results for the synthetic data with informed priors, randomly generated Bayes act, and Bayes error set to 0.2.

- In each case, we generate the synthetic data using a randomly generated $h_*$ with coordinates uniformly sampled in the interval $[0, 1]$. The reported results in Figures 4-6 are averages over 10 runs for each tested sample size.

Figure 6: Results for the synthetic data with informed priors, randomly generated Bayes act, and Bayes error set to 0.1.