[Reviews · NeurIPS 2019]

Reviewer 1



== SUMMARY == PAC-Bayesian risk bounds are considered by many to be the tightest existing analysis of generalization error. Despite this, there is still a gap between empirical test errors and analytic bounds, and this gap motivates further refinement of PAC-Bayes analysis. PAC-Bayes bounds come in various "flavors," the most popular one being the "little kl" bound, which bounds the KL divergence between Bernoulli distributions with parameters equal to the empirical risk and true risk, respectively. This form is the tightest, but considered by some as not easy to interpret. McAllester showed (COLT, 2003) how such bounds can be converted into a more interpretable bound on the difference of the empirical and true risks. One attractive facet of his bound is that when the empirical risk is small, the bound achieves the "fast" rate of O(1/n) -- thus capturing both realizable and non-realizable learning problems. Later, Tolstikhin & Seldin (TS; NIPS, 2013) derived a bound that accounts for the variance of the empirical losses. Since the empirical variance is often smaller than the empirical mean, their bound could be tighter than McAllester's (or others based on McAllester's analysis of the kl). Unfortunately, with all due respect to TS, their bound lacks the interpretability of McAllester's bound. (Of course, interpretability is subjective; this is simply this reviewer's opinion.) The current manuscript provides a bound that is similar in form to McAllester's, but includes a variance-like term. This term, denoted V_n in the paper, is the expected mean-squared error between a hypothesis drawn from the posterior and a reference hypothesis -- which could either be deterministic, or drawn from an alternate posterior -- trained on a subset of the data. To be precise, there are two such terms, arising from a split of the data. (Thanks to this splitting, the bound lends itself nicely to data-dependent priors, where you learn a prior on a subset of the data and use the rest to learn the posterior.) Unlike McAllester's bound, this one also has an irreducible term, V'_n, that in the worst case is O(1 / \sqrt{n}). The analysis follows from a modification of Bernstein's moment inequality where the X^2 variable is taken outside of its expectation -- hence the name, "un-expected Bernstein inequality." == PROS == The bound does appear to be tighter than existing bounds. The paper demonstrates this experimentally, which is helpful. It's not always easy to compare tightness just by looking at equations, so it's nice to see the claim supported by experiments. The form of the bound is, in my opinion, a bit more digestible than TS's bound. Though the variance-like quantity is not as interpretable as the sample variance, it is nice that the bound can be stated in one expression (i.e., without cases), and the paper gives an optimized form of the bound (Eq 1 combined with the Corollary of Theorem 1) which makes it easy to compare to existing bounds. == CONS == The irreducible term is disappointing. McAllester's bound does not have this term, so one would think it would be possible to remove it and achieve the fast rate. Perhaps the worst case upper bound on V'_n is too pessimistic? The experiments indicate that V'_n is on the order of the empirical risk, which could be O(1/n) under reasonable conditions. (McAllester's bound achieves the fast rate in this case.) While the bound seems to me to be more interpretable than TS's bound, it is not as interpretable as McAllester's. If you apply McAllester's kl analysis to a bound by Maurer, you get a bound that is, empirically, almost as tight as the current one, but more interpretable. If I were a practitioner looking for an "off-the-shelf" bound, I'd reach for the simpler one that is, practically speaking, about as tight. == PRESENTATION == The paper is very well written, which is much appreciated. The authors have taken care to make the theory accessible, starting with a generic bound in Eq 1, and a helpful corollary to interpret their main result. The presentation of the main theorem is a bit dense, but I suppose that is unavoidable. == DETAILED COMMENTS == Line 22: "Standard bounds all have an order ..." This is misleading. Most risk bounds have a term that is O(\sqrt{COMP_n / n}), but only a few have the empirical risk in that term. You can take any little-kl bound and convert it into a bound with that term, using McAllester's analysis. (I think there is also a risk bound due to Pollard with a similar term, but I can't find the reference.) The form of TS's empirical Bernstein bound is, as I understood it, much different from Eq 1. If the posterior satisfies a certain condition, you get the fast rate; if not, you get the variance-based term. I guess you could combine the terms with an indicator function of posterior -- is this what you mean? It would be helpful if you showed the TS bound in the form of Eq 1. (Apologies if this was somewhere in the appendix and I missed it.) How critical is data splitting to the bound? Is the bound meaningful if you take m = 0 (or m = n)? Lines 142 - 144: "... we could use an estimator that ... in the first term." This sentence was unclear to me. What are constants p and q in Eq 5? Are they used to split the data for cross-validation? How are p, q, m and n chosen in the experiments? Stability is mentioned in line 190, but it's not clear how stability plays into the bound. Why is it important that \hat{h}(Z_{m}) are close to \hat{h}(Z_{

Reviewer 2



The bound is a bit harder to deal with because it is formulated as an oracle inequality, and this introduces additional complexity in terms of tuning (quantization and union bound). The paper also includes some empirical evaluation, which I didn't find particularly strong. An interesting part here would be a synthetic evaluation, however it is somewhat "fixed", in a sense that it does not vary relevant parameters of the experiment to show sensitivity of the variance proxy (and it's merits against another empirical Bernstein type of bounds). For instance, this can be done by varying the Bayes optimal predictor. Therefore it is impossible to judge whether the bound is empirically tighter compared to the prior work. Also, it is unclear how far we are from the ground truth --- Bayes error in the plots would definitely help. Finally, comparison to another PAC-Bayesian literature that introduces algorithmic stability ideas is missing. For instance: Rivasplata, Omar, et al. "PAC-Bayes bounds for stable algorithms with instance-dependent priors." Advances in Neural Information Processing Systems. 2018. == Post-rebuttal comments I would like to thank authors for their detailed response. Promised improvements will definitely make the paper better. There was also a number of issues raised during the discussion (see comments by the meta-reviewer). I agree that at the moment, the paper perhaps has too many "moving parts" and their effect should be ideally studied separately (that is, the effect of the "half-samples", biasing, new Bernstein-style bound). The work would be much more solid if this would be the case. If the paper is not accepted at this time, this is the main point for improvement. As for the conventions, I also agree with the meta-reviewer that "un-expected" indeed sounds a bit strange, and this could be changed regardless of acceptance.

Reviewer 3



Originality: The paper provides a new PAC-Bayesian bound based on a notion of stability of the learning algorithm on two subsets of the sample. While the authors discuss links with other PAC-Bayesian bounds including Tolstikhin and Seldin's bound based on a different version of Bernstein's inequality, they fail to cite or compare their approach to Rivasplata et al. (NeurIPS 2018), PAC-Bayes bounds for stable algorithms with instance-dependent priors. This paper also combines the PAC-Bayes approach with the stability of the hypothesis learned by a learning algorithm on the sample, and also state that the empirical values can be much smaller than previous state-of-the-art PAC-Bayesian bounds. While I did not thoroughly compare the proposed bound with Rivasplata's bound, I feel that the paper can not be accepted unless the authors do such a comparison, in terms of theoretical properties and empirical results. Quality: The paper is complete and seem technically sound - all results are presented with complete proofs (and even a new notation to simplify some theorems and proofs). Empirical results on a toy dataset and some UCI datasets are provided to show the advantage of the bound when compared to other PAC-Bayesian bounds of the literature. Quality: Sections 1 to 4 are clearly written and are easy to follow even if the subject at hand is complex. However, the following sections leave me confused - there are many back-and-forth discussions and results that I feel should have been introduced earlier in the paper, giving me the impression that this paper was originally shorter with a bigger appendix with complete results. My conclusion on this matter is that I think the paper could be easier to follow if it was more polished (and possibly reorganized). Significance: The new Bernstein inequality seem significant and could be reused by other researches. However, the lack of comparison with Rivasplata's NeurIPS 2018 paper makes it hard to make a decision on the significance of the new PAC-Bayes bound or the authors.

[Author Response · NeurIPS 2019]

**@Reviewer 1.** Your detailed comments were much appreciated. ● The irreducible term involving $V_n'$ does not
necessarily prevent a fast rate in the realizable setting as we will clarify now: First, suppose that $Q$ is chosen to satisfy (6)
(this assumption will simplify the argument that follows at the price of weakening our bound slightly). With this choice
of $Q$, one can now see that $V_n'$ is just a measure of the average performance of hypotheses drawn from the "alternate
posterior" $Q_{\leq m}$ [resp. $Q_{>m}$] on the *unseen* i.i.d. sample $Z_{>m}$ [resp. $Z_{\leq m}$]. Thus, when $|\ell| \leq 1$ and $m = n/2$, the first
sum in the expression of $V_n'$ satisfies $\sum_{j=1}^{m} \mathbb{E}_{h \sim Q_{>m}}[\ell_h(Z_j)^2]/n \leq \sum_{j=1}^{m} \mathbb{E}_{h \sim Q_{>m}}[\ell_h(Z_j)]/n \leq L(Q_{>m})/2 + \tilde{O}(1/\sqrt{n})$,
where the last inequality holds w.h.p. due to Hoeffding. By applying the same treatment to the second sum in $V_n'$, we
arrive at (*): $V_n' \leq (L(Q_{\leq m}) + L(Q_{>m}))/2 + \tilde{O}(1/\sqrt{n})$ (w.h.p.). If $Q$ is chosen to be ERM, for example, then the risks
$L(Q_{\leq m})$ and $L(Q_{>m})$ are often very small in the realizable setting (as they approach the zero Bayes risk). If these risks
are of order $1/\sqrt{n}$, then due to (*), our irreducible term $\mathcal{C}\sqrt{R_n' \cdot s_{\delta,n}/n} = \mathcal{C}\sqrt{V_n' \cdot s_{\delta,n}/n}$ in (1) is of order $\tilde{O}(1/n)$. In
contrast, McAllester's bound can be of order $\tilde{O}(\mathrm{KL}/n)$ in the realizable setting, and so in this case our irreducible term
is of comparable size or even smaller since it is "KL"-free. The same conclusion can be drawn in the *non-realizable*
setting, since in this case, for $n$ larger enough, $\sqrt{L_n(P_n) \cdot \mathrm{KL}/n}$ becomes the dominant term in McAllester's bound
which, again due to the KL and (*) above, can be larger than our irreducible term. The argument above should also
answer your question about whether the upper-bound $V_n' \leq b^2$ (or the $O(b/\sqrt{n})$ upper-bound on the irreducible term) is
too pessimistic—indeed it is as revealed by (*) above. ● Though it may be more interpretable, McAllester's bound,
which is the result of the $\mathrm{kl}$ analysis of Maurer's bound (see, e.g., (3) in TS paper), is far from being competitive with
our bound—in our experiments it performs worse than Catoni's bound; the bound due to Maurer that we report on in our
experiments is exactly (2) without any relaxation (see lines 152-153), which is much tighter than McAllester's—this is
also why we only reported results for the former. ● We will modify line 22 to reflect your point. ● From (*), one would
expect $m = n/2$ to be the optimal choice for the data splitting. However, with the convention that $Q(\varnothing)$ equals some
prior, say $P_0$, the bound would still be meaningful for $m \in \{0, n\}$. ● It can be shown that TS's bound can equivalently
be written as (constants and log-factors omitted): $L(P_n) - L_n(P_n) \leq \max(\sqrt{\mathbb{V}_n \cdot \mathrm{KL}/n}, \mathrm{KL}/n)$, where $\mathbb{V}_n$ is the
empirical loss variance—the case distinction they consider is merely another way of writing the same inequality. This
inequality can be relaxed using $\max(a, b) \leq a + b, \forall a, b > 0$, to recover a bound of the form (1). We will explain this
point in the appendix and point to it in Sec. 2. ● Lines 142-144: Consider the second term in $V_n$, in the Corollary of
Thm. 1, involving the sum from $j = m + 1$ to $n$; looking inside the square, we see that to predict the data point $Z_j$, the
estimator $\hat{h}$ only uses the sample $Z_{\leq m}$ and suffers loss $\ell_{\hat{h}(Z_{\leq m})}(Z_j)$. In contrast, in Thm. 1, one can use the posterior
$Q_{<j} \equiv \delta(\hat{h}(Z_{<j}))$ which depends on the extra points $Z_{m+1}, \ldots, Z_{j-1}$ and suffers loss $\ell(\hat{h}(Z_{<j}), Z_j)$. This loss gets
closer and closer (as $j \to n$) to the loss that would be incurred by the estimator $\hat{h}(Z_{\leq n})$ "trained" on the whole sample
$Z_{\leq n}$. ● In the experiments, $m = n/2$ and in (5) we considered $(p, q)$ equal to $(1, m), (m + 1, n)$, and $(1, n)$, to compute
$Q_{\leq m} \equiv \delta(\hat{h}(Z_{\leq m})), Q_{>m} \equiv \delta(\hat{h}(Z_{>m}))$, and $\hat{h}(Z_{\leq n})$ (the latter is used for $P_n$). ● In our experiments the posterior
$P_n$ is a Gaussian centered at $\hat{h}(Z_{\leq n})$. So, to see why it is important for $\hat{h}(Z_{\leq m})$ and $\hat{h}(Z_{>m})$ to be close to $\hat{h}(Z_{\leq n})$,
consider the extreme case where $P_n$ has zero variance, i.e. $P_n \equiv \delta(\hat{h}(Z_{\leq n}))$. In this case, it is clear from the expression
of $V_n$ that, if $\hat{h}(Z_{\leq m}) = \hat{h}(Z_{>m}) = \hat{h}(Z_{\leq n})$, then $V_n = 0$. So, when $P_n$ has non-zero but small enough variance, one
would still expect $V_n \simeq 0$, when $\hat{h}(Z_{\leq m}) \simeq \hat{h}(Z_{>m}) \simeq \hat{h}(Z_{\leq n})$, which can make the first term on the RHS of (1) small.

**@Reviewer 2.** Thank you for your feedback on the experiments. ● When running the synthetic experiments, we
found that the bounds were highly sensitive to one particular parameter—the variance of the Gaussian posterior $P_n$. For
this reason, the variance was optimized for every bound separately (see lines 165-167). The sensitivity w.r.t. the Bayes
Optimal Predictor (BOP) was weak; varying the BOP did not change the relative ranking of the bounds or affect the gap
between them by much, and so due to space we did not include results for different BOPs. Nevertheless, we will now
add the cases where the Bayes error is equal to 0.05 and 0.2 for randomly generated BOPs in the appendix. ● In Figure
1, the Bayes error is 0.1 (the true labels are flipped with probability $1 - 0.9$, see line 173); we will make this clearer by
adding it to the Figure directly. ● We will cite and briefly discuss Rivasplata et al.'s paper in the relevant section.

**@Reviewer 3.** Thank you for pointing out Rivasplata et al.'s paper. ● Although their title might suggest otherwise,
the ideas, techniques, and results of their paper are substantially different from ours. In fact, even though the title of their
paper mentions "instance-based", to compute their bound, one needs to know the uniform (worst case over *all* possible
samples, not just the observed one) stability for the learning algorithm involved. For many popular algorithms, such as
gradient descent for example, no non-trivial bound on this worst-case stability parameter is known. In contrast, we can
get non-trivial bounds for *any* algorithm as soon as our *empirical* notion of stability $V_n$—which can be calculated on
the data—is small. We will cite Rivasplata et al. and explain this difference in the relevant section. ● The experiments
in our paper are based on $\lambda$-penalized logistic regression, which happens to be an algorithm for which Rivasplata's
worst-case stability parameter $\beta_n$ can be calculated after all, giving $\beta_n = 1/(\lambda n)$. We tested Rivasplata's bound with
this value of $\beta_n$, and found that in our experiments it performs worse than the bounds we currently compare ours against
(i.e. Catoni, Maurer, and TS). For completeness, we will add these additional results with discussion in the appendix.

[Meta-Review · NeurIPS 2019]

The paper introduces three exciting ideas to the area of PAC-Bayesian analysis: (1) a new way of using "half-samples" to construct informed priors; (2) offsetting (biasing) the loss estimate by the loss of a reference hypothesis h_* to achieve "fast convergence rates" under Bernstein condition [even when the loss itself is bounded away from zero]; (3) a new form of Empirical Bernstein inequality, which is combined with PAC-Bayes to exploit low variance [the need in a new inequality and its advantages are not well explained]. The authors compare a bound based on combination of the three ideas with PAC-Bayes bound of Maurer (2004) and some other PAC-Bayes bounds, demonstrating superiority of the new approach. While the work is really exciting, the authors fail to clearly separate between the three major contributions. It is not shown how much each of the three novelties contribute to the success of the method. [I believe it's the biasing giving the most, rather than the new Empirical Bernstein. Biasing and informed priors can be easily combined with the bound of Tolstikhin & Seldin (2013) [TS] and this comparison should be added.] The relation of the new results to prior work is not sufficiently explored and the key ideas could be explained in a much simpler and cleaner way. And, please, do not call your bound Un-Expected Bernstein - see my explanation below. I have the following suggestions for improvement: Regarding informed priors: 1. Add a simple explanation of the idea of using "half-samples" for construction of informed priors and how it can be combined with existing bounds. I provide a one-line explanation below. 2. Add a comparison of Maurer's bound with uninformed prior, KL(P_n||P_0), with Maurer's bound with informed prior, KL(P_n||P_< m) + KL(P_n||P_\geq m). 3. Add a comparison of Maurer's bound with informed prior with the new version of PAC-Bayes Empirical Bernstein. 4. Ambroladze et al. use the first half of a sample to construct a prior, which is used for the second half of a sample. Let's call it a "forward" approach. Your construction is equivalent to taking an average of "forward" and "backward" approaches. It is elegant and more stable, but on average it is the same. [Therefore, I believe the improvement does not come from here.] Regarding biasing: 5. Add a simple explanation of the idea. I provide a one-line explanation below. 6. Biasing can be easily combined with PAC-Bayes Empirical Bernstein bound of TS [see the derivation below] and the need in derivation of a new bound and its advantages are not explained. I see why it is problematic to use informed priors in the final bound of TS, because of the square root, but I believe it is possible in the intermediate linear version of the bound in equation (15) in TS, see the derivation. And even if not, you can train the prior and bias on the first half of the sample and apply the bound of TS to the second half. As explained above, on average this is equivalent to what you obtain from your "forward" and "backward" split. This comparison should be added. Regarding the new form of Empirical Bernstein bound: 7. Calling the new bound "un-expected Bernstein" is very confusing. Replacing expected values with empirical ones is the main idea of Empirical Bernstein. The fact that you take X^2 out of the expectation is not what distinguishes your result from existing Empirical Bernstein inequalities (and there are several of them). All of them could be called "Un-expected Bernstein", but they were called "Empirical Bernstein" and I urge you to keep with the tradition. Otherwise, I can easily see people getting confused, starting rediscovering things, not comparing with the right baselines, missing relevant prior work, etc. Please, don't do that! 8. The comparison of the new Empirical Bernstein inequality with Maurer & Pontil (2009) should move from the appendix to the body of the paper. In the appendix you show that a relaxation of the new bound is weaker than Maurer & Pontil. So what are the advantages of the new bound? How they compare numerically? [It does not feel like the improvement is coming from here.] === Below I provide a one-line explanation of the idea of informed priors and how they can be incorporated into existing PAC-Bayes bounds, specifically Maurer's bound and PAC-Bayes bounds in a linear form. Then a one-line explanation of biasing and how it could be combined with the bound of TS. Then I give some minor comments and references. === One-line explanation of informed priors: Let L_1(h) be the empirical loss of h on the first half of the sample and L_2(h) on the second half. We then have that L_n(h) = (1/2) L_1(h) + (1/2) L_2(h) is the empirical loss on the full sample. Let P_1 be the prior constructed on the first half of the sample and P_2 on the second half. Let P_n be the posterior. Let E[L] = E_{h ~ P_n}[L(h)] be the expected loss when h is drawn according to P_n. Let E[L_1] = E_{h ~ P_n}[L_1(h)] and the same for E[L_2] and E[L_n]. Let X = \ln(2\sqrt{n}/\delta) be the logarithmic part of Maurer's bound. Then we have: kl(E[L]||E[L_n]) = kl((1/2)E[L] + (1/2)E[L]|| (1/2) E[L_1] + (1/2) E[L_2]) \leq (1/2) kl(E[L]||E[L_1]) + (1/2) kl(E[L]||E[L_2]) \leq (1/2) (KL(P_n||P_2) + X) / (n/2) + (1/2) (KL(P_n||P_1) + X) / (n/2) = (KL(P_n||P_2) + KL(P_n||P_1) + 2X) / n. Where: the first inequality is by Jensen's inequality and convexity of kl and the second is the Mauer's PAC-Bayes inequality. Maurer's bound is applicable, because the priors are built on a complimentary part of the sample, not the one used for estimating the empirical loss. The same kind of derivation can be applied to PAC-Bayes bounds in a linear form. === One-line explanation of biasing: Instead of bounding L(h) - L_n(h) directly, which is the standard way, you take a hypothesis h_*, which was trained on the other half of the sample and write L(h) - L_n(h) = L(h) - L(h_*) + L(h_*) - L_n(h_*) + L_n(h_*) - L_n(h). Then you use Empirical Bernstein (or PAC-Bayes Empirical Bernstein for distributions over H and you could use the bound of TS) to bound L(h) - L(h_*) + L_n(h_*) - L_n(h) and, for example, the kl bound to bound L(h_*) - L_n(h_*). The latter gives what you call the "irreducible term" (you work with just one hypothesis, which could also be a "randomized classifier" defined by a distribution over H). Strictly speaking, this term is not part of the PAC-Bayes bound: PAC-Bayes is used to bound "L(h) - L(h_*) + L_n(h_*) - L_n(h)" and a concentration inequality for a single hypothesis is used to bound "L(h_*) - L_n(h_*)". If you combine biasing with the bound of TS you would get the same separation. The term "L(h) - L(h_*) + L_n(h_*) - L_n(h)" concentrates well when there is a high degree of agreement in predictions of h and h_* and thus the variance is small. === Minor comments and references: Empirical Bernstein inequality has been introduced in Audibert, Munos, & Czepesvari, "Tuning bandit algorithms in stochastic environments.", ALT, 2007 and the name "Empirical Bernstein" was coined in Mnih, Czepesvari, & Audibert "Empirical Bernstein Stopping", ICML, 2008. I think these works deserve credit. An alternative form of Empirical Bernstein appears in Wintenberger, Optimal learning with Bernstein Online Aggregation, Machine Learning, 2017, based on an inequality from Cesa-Bianchi, Mansour, and Stoltz, Improved second-order bounds for prediction with expert advice, Machine Learning, 2007. The idea of applying PAC-Bayes to a cross-validation-type split of the data (including overlaps) appears in Thiemann, Igel, Wintenberger, and Seldin, A strongly quasiconvex PAC-Bayesian bound, ALT, 2017. Multiple overlapping splits could be directly used in your work to further improve stability. Your proof of change-of-measure inequality (Proposition 8) is actually very standard. I am not sure whether the use of ESI notation is beneficial. It's nice, but it makes things harder to follow for people who are not used to it.